# Learned Index with Dynamic $\epsilon$

## Abstract

Index structure is a fundamental component in database and facilitates broad data retrieval applications. Recent learned index methods show superior performance by learning hidden yet useful data distribution with the help of machine learning, and provide a guarantee that the prediction error is no more than a pre-defined $\epsilon$. However, existing learned index methods adopt a fixed $\epsilon$ for all the learned segments, neglecting the diverse characteristics of different data localities. In this paper, we propose a mathematically-grounded learned index framework with dynamic $\epsilon$, which is efficient and pluggable to existing learned index methods. We theoretically analyze prediction error bounds that link $\epsilon$ with data characteristics for an illustrative learned index method. Under the guidance of the derived bounds, we learn how to vary $\epsilon$ and improve the index performance with a better space-time trade-off. Experiments with real-world datasets and several state-of-the-art methods demonstrate the efficiency, effectiveness and usability of the proposed framework.

## 1 Introduction

Data indexing [15, 32, 22, 34], which stores keys and corresponding payloads with designed structures, supports efficient query operations over data and benefits various data retrieval applications. Recently, Machine Learning (ML) models have been incorporated into the design of index structure, leading to substantial improvements in terms of both storage space and querying efficiency [17, 11, 24, 31]. The key insight behind this trending topic of "learned index" is that the data to be indexed contain useful distribution information and such information can be utilized by trainable ML models that map the keys $\{x\}$ to their stored positions $\{y\}$.

To approximate the data distribution, state-of-the-art (SOTA) learned index methods [14, 18, 12, 10] propose to learn piece-wise linear segments $\mathbf{S} = [S_1, ..., S_i, ..., S_N]$, where $S_i : y = a_i x + b_i$ is the linear segment parameterized by $(a_i, b_i)$ and $N$ is the total number of learned segments. These methods introduce an important pre-defined parameter $\epsilon \in \mathbb{Z}_{>1}$ and adopt the following online learning process: Beginning from the first available data point, the current linear segment adjusts $(a_i, b_i)$ and covers as many data points as possible until a data point, say $(x', y')$ achieves the prediction error $|S_i(x') - y'| > \epsilon$. The violation of $\epsilon$ triggers a new linear segment, and the data point $(x', y')$ will be the first available data point. The process repeats until no data point is available and as a result, the worst-case preciseness can be guaranteed with $\epsilon$.

By tuning $\epsilon$, various space-time preferences from users can be met. For example, a relatively large $\epsilon$ can result in a small index size while having large prediction errors, and on the other hand, a relatively small $\epsilon$ provides users with small prediction errors while having more learned segments and thus a large index size. However, existing learned index methods implicitly assume that the whole dataset to be indexed contains the same characteristics for different localities and thus adopt the same $\epsilon$ for all the learned segments, leading to sub-optimal index performance. More importantly, the impact of $\epsilon$ on index performance is intrinsically linked to data characteristics, which are not fully explored and utilized by existing learned index methods.

Motivated by these, in this paper, we theoretically analyze the impact of $\epsilon$ on index performance, and link the characteristics of data localities with the dynamic adjustments of $\epsilon$. Based on the derived

theoretical results, we propose an efficient and pluggable learned index framework that dynamically adjusts $\epsilon$ in a principled way. To be specific, under the setting of an illustrative learned index method MET [10], we present novel analysis about the prediction error bounds of each segment that link $\epsilon$ with the mean and variance of data localities. The segment-wise prediction error embeds the space-time trade-off as it is the product of the *number of covered keys* and *mean absolute error*, which determine the index size and preciseness respectively. The derived mathematical relationships enable our framework to fully explore diverse data localities with an $\epsilon$-learner module, which learns to predict the impact of $\epsilon$ on the index performance and adaptively choose a suitable $\epsilon$ to achieve a better space-time trade-off.

We apply the proposed framework to several SOTA learned index methods, and conduct a series of experiments on three widely adopted real-world datasets. Comparing with the original learned index methods with fixed $\epsilon$, our dynamic $\epsilon$ versions achieve significant index performance improvements with better space-time trade-offs. We also conduct various experiments to verify the necessity and effectiveness of the proposed framework, and provide both ablation study and case study to understand how the proposed framework works. Our contributions can be summarized as follows:

- We make the first step to exploit the potential of dynamically adjusting $\epsilon$ for learned indexes, and propose an efficient and pluggable framework that can be applied to a broad class of piece-wise approximation algorithms.
- We provide theoretical analysis for a proxy task modeling the index space-time trade-off, which establishes our $\epsilon$-learner based on the data characteristics and the derived bounds.
- We achieve significant index performance improvements over several SOTA learned index methods on real-world datasets. To facilitate further studies, we make our codes and datasets public at https://github.com/AnonyResearcher/NeurIPS-5930.

## 2   Background

**Learned Index.**   Given a dataset $\mathcal{D} = \{(x,y)|x \in \mathcal{X}, y \in \mathcal{Y}\}$, $\mathcal{X}$ is the set of *keys* over a universe $\mathcal{U}$ such as reals or integers, and $\mathcal{Y}$ is the set of *positions* where the keys and corresponding payloads are stored. The index such as B$^+$-tree [1] aims to build a compact structure to support efficient query operations over $\mathcal{D}$. Typically, the keys are assumed to be sorted in ascending order to satisfy the *key-position monotonicity*, *i.e.*, for any two keys, $x_i > x_j$ iff their positions $y_i > y_j$, such that the range query ($\mathcal{X} \cap [x_{low}, x_{high}]$) can be handled.

Recently, learned index methods [19, 20, 30, 7, 6] leverage ML models to mine useful distribution information from $\mathcal{D}$, and incorporate such information to boost the index performance. To look up a given key $x$, the learned index first predicts position $\hat{y}$ using the learned models, and subsequently finds the stored true position $y$ based on $\hat{y}$ with a binary search or exponential search. Thus the querying time consists of the inference time of the learned models and the search time in $O(\log(|\hat{y} - y|))$. By modeling the data distribution information, learned indexes achieve faster query speed than traditional B$^+$-tree index, meanwhile using several orders-of-magnitude smaller storage space [9, 14, 12, 18, 23].

**$\epsilon$-bounded Linear Approximation.**   Many existing learned index methods adopt piece-wise linear segments to approximate the distribution of $\mathcal{D}$ due to their effectiveness and low computing cost, and introduce the parameter $\epsilon$ to provide a worst-case preciseness guarantee and a tunable knob to meet various space-time trade-off preferences. Here we briefly introduce the SOTA $\epsilon$-bounded learned index methods that are most closely to our work, and refer to the review chapter of [11] for details of other methods. We first describe an illustrative learned index algorithm MET [10]. Specifically, for any two consecutive keys of $\mathcal{D}$, suppose their key interval $(x_i - x_{i-1})$ is drawn according to a random process $\{G_i\}_{i \in \mathbb{N}}$, where $G_i$ is a positive independent and identically distributed (i.i.d.) random variable whose mean is $\mu$ and variance is $\sigma^2$. MET learns linear segments $\{S_i : y = a_i x + b_i\}$ via a simple deterministic strategy: the current segment fixes the slope $a_i = 1/\mu$, goes through the first available data point and thus $b_i$ is determined. Then $S_i$ covers the remaining data points one by one until a data point $(x', y')$ gains the prediction error larger than $\epsilon$. The violation triggers a new linear segment that begins from $(x', y')$ and the process repeats until $\mathcal{D}$ has been traversed.

Other $\epsilon$-bounded learned index methods learn linear segments in a similar manner to MET while having different mechanisms to determine the parameters of $\{S_i\}$. FITing-Tree [14] uses a greedy shrinking cone algorithm. PGM [12] adopts another one-pass algorithm that achieves the optimal number of learned segments. Radix-Spline [18] introduces a radix structure to organize the learned segments. However, existing methods constrain all learned segments with the same $\epsilon$. All of these piece-wise segments based approaches attempt to improve performance by changing the way

98 segments are learned or organized, but ignore the optimization potential of dynamically varying $\epsilon$.
99 In this paper, we will discuss the impact of $\epsilon$ in more depth and investigate how to enhance existing
100 learned index methods from a new perspective: dynamic adjustment of $\epsilon$ accounting for the diversity
101 of different data localities. Besides, different from [10] that reveals the relationship between $\epsilon$ and
102 index size performance based on MET. In Section 3.3, we present novel analysis about the impact of
103 $\epsilon$ on not only the index size, *but also the index preciseness and a comprehensive trade-off quantity*,
104 which facilitates the proposed dynamic $\epsilon$ adjustment.

# 3 Learn to Vary $\epsilon$

## 3.1 Problem Formulation and Motivation

107 Before introducing the proposed framework, we first formulate the task of learning index from
108 data with $\epsilon$ guarantee, and provide some discussions about why we need to vary $\epsilon$. Given a dataset
109 $\mathcal{D}$ to be indexed and an $\epsilon$-bounded learned index algorithm $\mathcal{A}$, we aim to learn linear segments
110 $\mathbf{S} = [S_1, ..., S_i..., S_N]$ with segment-wise varied $[\epsilon_i]_{i \in [N]}$, such that a better trade-off between
111 storage cost (size in KB) and query efficiency (time in ns) can be achieved than the ones using fixed
112 $\epsilon$. Let $\mathcal{D}_i \subset \mathcal{D}$ be the data whose keys are covered by $S_i$, for the remaining data $\mathcal{D} \setminus \bigcup_{j<i} \mathcal{D}_j$, the
113 algorithm $\mathcal{A}$ repeatedly checks whether the prediction error of new data point violates the given $\epsilon_i$
114 and outputs the learned segment $S_i$. When all the $\epsilon_i$s for $i \in [N]$ take the same value, the problem
115 becomes the one that existing learned index methods are dealing with.

116 To facilitate theoretical analysis, we focus on two proxy quantities for the target space-time trade-off:
117 (1) the number of learned segments $N$ and (2) the mean absolute prediction error $MAE(\mathcal{D}_i|S_i)$, which
118 is affected and upper-bounded by $\epsilon_i$. We note that the improvements of $N$-$MAE$ trade-off fairly and
119 adequately reflect the improvements of the space-time trade-off: (1) The learned segments size in
120 bytes and $N$ are positively correlated and only different by a constant factor, *e.g.*, the size of a segment
121 can be 128bit if it consists of two double-precision float parameters (slope and intercept); (2) When
122 using exponential search, the querying complexity is $O(\log(N) + \log(MAE(\mathcal{D}_i|S_i)))$, in which the
123 first term indicates the finding process of the specific segment $S'$ that covers the key $x$ for a queried
124 data point $(x, y)$, and the second term indicates the search range $|\hat{y} - y|$ for true position $y$ based on
125 the estimated one $\hat{y} = S'(x)$. In this paper, we adopt exponential search as search algorithm since it
126 is better than binary search for *exploiting the predictive ability of learned models*. In Appendix C,
127 we show that the search range of exponential search is $O(MAE(\mathcal{D}_i|S_i))$, which can be much smaller
128 than the one of binary search, $O(\epsilon_i)$, especially for strong predictive models and the datasets having
129 clear linearity. Similar empirical support can be also found in [9].

130 Now let's examine how the parameter $\epsilon$ affects the $N$-$MAE$ trade-off. We can see that these two
131 performance terms compete with each other and $\epsilon$ plays an important role to balance them. If we
132 adopt a small $\epsilon$, the prediction error constraint is more frequently violated, leading to a large $N$;
133 meanwhile, the preciseness of learned index is improved, leading to a small *MAE* of the whole data
134 $MAE(\mathcal{D}|\mathbf{S})$. On the other hand, with a large $\epsilon$, we will get a more compact learned index (*i.e.*, a small
135 $N$) with larger prediction errors (*i.e.*, a large $MAE(\mathcal{D}|\mathbf{S})$).

136 Actually, the effect of $\epsilon$ on index performance is intrinsically linked to the characteristic of the
137 data to be indexed. For real-world datasets, an important observation is that *the linearity degree*
138 *varies in different data localities*. Recall that we use piece-wise linear segments to fit the data, and
139 $\epsilon$ determines the partition and the fitness of the segments. By varying $\epsilon$, we can adapt to the local
140 variations of $\mathcal{D}$ and adjust the partition such that each learned segment fits the data better. Formally,
141 let's consider the quantity $SegErr_i$ that is defined as the total prediction error within a segment $S_i$,
142 *i.e.*, $SegErr_i \triangleq \sum_{(x,y) \in \mathcal{D}_i} |y - S_i(x)|$, which is also the product of the number of covered keys
143 $Len(\mathcal{D}_i)$ and the mean absolute error $MAE(\mathcal{D}_i|S_i)$. Note that a large $Len(\mathcal{D}_i)$ leads to a small $N$
144 since $|\mathcal{D}| = \sum_{i=1}^{N} Len(\mathcal{D}_i)$. From this view, the quantity $SegErr_i$ internally reflects the $N$-$MAE$
145 trade-off. Later we will show how to leverage this quantity to dynamically adjust $\epsilon$.

## 3.2 Overall Framework

147 In practice, it is intractable to directly solve the problem formulated in Section 3.1. With a given $\epsilon_i$,
148 the one-pass algorithm $\mathcal{A}$ determines $S_i$ and $\mathcal{D}_i$ until the error bound $\epsilon_i$ is violated. In other words,
149 it is unknown what the data partition $\{\mathcal{D}_i\}$ will be *a priori*, which makes it impossible to solve the
150 problem by searching among all the possible $\{\epsilon_i\}$s and learning index with a set of given $\{\epsilon_i\}$.

151 In this paper, we investigate how to efficiently find an approximate solution to this problem via the
152 introduced $\epsilon$-*learner* module. Instead of heuristically adjusting $\epsilon$, the $\epsilon$-*learner* learns to predict

the impact of $\epsilon$ on the index structure and adaptively adjusts $\epsilon$ in a principled way. Meanwhile, the
introducing of *$\epsilon$-learner* should not sacrifice the efficiency of the original one-pass learned index
algorithms, which is important for real-world practical applications.

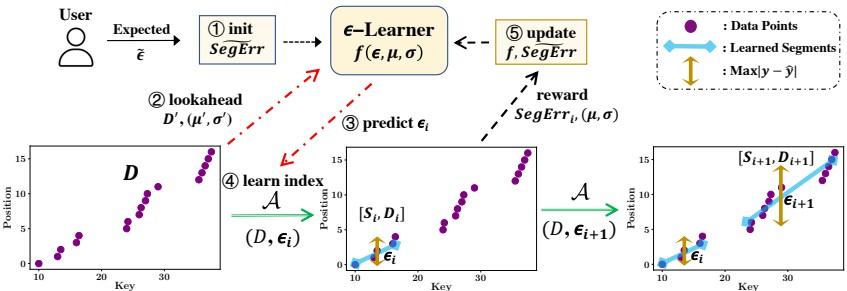

Figure 1: The dynamic $\epsilon$ framework. We ① transform $\tilde{\epsilon}$ into the proxy prediction error $\widetilde{SegErr}$, then
② sample a small look-ahead data $D'$ to estimate the data characteristics $(\mu, \sigma)$. ③ The $\epsilon$-learner
predicts a suitable $\epsilon_i$ accordingly, and ④ we learn a new segment $S_i$ using $\mathcal{A}$ (*e.g.*, PGM) with $\epsilon_i$.
⑤ Once $S_i$ triggers the violation of $\epsilon_i$, the $\epsilon$-learner is updated and enhanced with the rewarded
ground-truth. Steps ② to ⑤ repeat in an online manner to approximate the distribution of $\mathcal{D}$.

These two design considerations establish our dynamic $\epsilon$ framework as shown in Figure 1. The
*$\epsilon$-learner* is based on an estimation function $SegErr = f(\epsilon, \mu, \sigma)$ that depicts the mathematical
relationships among $\epsilon$, $SegErr_i$ and the characteristics $\mu, \sigma$ of the data to be indexed. As a start, users
can provide an expected $\tilde{\epsilon}$ that indicates various preferences under space-sensitive or time-sensitive
applications. To meet the user requirements, afterwards, we internally transform the $\tilde{\epsilon}$ into another
proxy quantity $\widetilde{SegErr}$, which reflects the expected prediction error for each segment if we set $\epsilon_i = \tilde{\epsilon}$.
This transformation also links the adjustment of $\epsilon$ and data characteristics together, which enables the
data-dependent adjustment of $\epsilon$. Beginning with $\tilde{\epsilon}$, the *$\epsilon$-learner* chooses a suitable $\epsilon_i$ according to
current data characteristics, then learns a segment $S_i$ using $\mathcal{A}$, and finally enhances the *$\epsilon$-learner* with
the rewarded ground-truth $SegErr_i$ of each segment. To make the introduced adjustment efficient,
we propose to only sample a small *Look-ahead* data $\mathcal{D}'$ to estimate the characteristics $(\mu, \sigma)$ of the
following data locality. The learning process repeats and is also in an efficient one-pass manner.

Note that the proposed framework provides users the same interface as the ones used by original
learned index methods. That is, we do not add any additional cost to the users' experience, and users
can smoothly and painlessly use our framework with given $\tilde{\epsilon}$ just as they use the original methods
with given $\epsilon$. The $\epsilon$ is an intuitive, meaningful, easy-to-set and method-agnostic quantity for users.
On the one hand, we can easily impose restrictions on the worst-case querying cases with $\epsilon$ as the data
accessing number in querying process is $O(\log(|\hat{y} - y|))$. On the other hand, $\epsilon$ is easier to estimate
than the other quantities such as index size and querying time, which are dependent on specific
algorithms, data layouts, implementations and experimental platforms. Our pluggable framework
retains the benefits of existing learned index methods, such as the aforementioned usability of $\epsilon$, and
the ability to handle dynamic update case and hard size requirement. [1]

We have seen how $\epsilon$ determines index performance and how $SegErr_i$ embeds the *N-MAE* trade-off
in Section 3.1. In Section 3.3, we further theoretically analyze the relationship among $\epsilon$, $SegErr_i$,
and data characteristics $\mu, \sigma$ at different localities. Based on the analysis, we elaborate the details of
*$\epsilon$-learner* and the internal transformation between $\epsilon$ and $SegErr_i$ in Section 3.4.

## 3.3 Prediction Error Estimation

In this section, we theoretically study the impact of $\epsilon$ on the prediction error $SegErr_i$ of each learned
segment $S_i$. The derived closed-form relationships will be taken into account in the design of the
proposed $\epsilon$-learner module (Section 3.4). Specifically, for the MET algorithm, we can prove the
following theorem to bound the expectation of $SegErr_i$ with $\epsilon$ and the key interval distribution of $\mathcal{D}$.

**Theorem 1.** *Given a dataset $\mathcal{D}$ to be indexed and an $\epsilon$ where $\epsilon \in \mathbb{Z}_{>1}$, consider the setting of the*
*MET algorithm [10], in which key intervals of $\mathcal{D}$ are drawn from a random process consisting of*
*positive i.i.d. random variables with mean $\mu$ and variance $\sigma^2$, and $\epsilon \gg \sigma/\mu$. For a learned segment*
*$S_i$ and its covered data $\mathcal{D}_i$, denote $SegErr_i = \sum_{(x,y)\in D_i} |y - S_i(x)|$. Then the expectation of*

---

[1]We discuss how to extend existing works in more details in Appendix E.

$SegErr_i$ satisfies:

$$\sqrt{\frac{1}{\pi}\frac{\mu}{\sigma}}\epsilon^2 < \mathbb{E}[SegErr_i] < \frac{2}{3}\sqrt{\frac{2}{\pi}}(\frac{5}{3})^{\frac{3}{4}}(\frac{\mu}{\sigma})^2\epsilon^3.$$

This theorem reveals that the prediction error $SegErr_i$ depends on both $\epsilon$ and the data characteristics $(\mu, \sigma)$. Recall that $CV=\sigma/\mu$ is the *coefficient of variation*, a classical statistical measure of the relative dispersion of data points. In the context of the linear approximation, the data statistic $1/CV = \mu/\sigma$ in our bounds intrinsically corresponds to the linearity degree of the data. With this, we can find that when $\mu/\sigma$ is large, the data is easy-to-fit with linear segments, and thus we can choose a small $\epsilon$ to achieve precise predictions. On the other hand, when $\mu/\sigma$ is small, it becomes harder to fit the data using a linear segment, and thus $\epsilon$ should be increased to absorb some non-linear data localities. In this way, we can make the total prediction error for different learned segments consistent and achieve a better N-*MAE* trade-off. This analysis also confirms the motivation of varying $\epsilon$: The local linearity degrees of the indexed data can be diverse, and we should adjust $\epsilon$ according to the local characteristic of the data, such that the learned index can fit and leverage the data distribution better.

In the rest of this section, we provide a proof sketch of this theorem due to the space limitation. For detailed proof, please refer to our Appendix A. The main idea is to model the learning process of linear approximation with $\epsilon$ guarantee as a random walk process, and consider that the absolute prediction error of each data point follows folded normal distributions. Specifically, given a learned segment $S_i : y = a_i x + b_i$, we can calculate the expectation of $SegErr_i$ for this segment as:

$$\mathbb{E}[SegErr_i] = a_i\mathbb{E}\left[\sum_{j=0}^{(j^*-1)}|Z_j|\right] = a_i\sum_{n=1}^{\infty}\mathbb{E}\left[\sum_{j=0}^{n-1}|Z_j|\right]\Pr(j^*=n), \qquad (1)$$

where $Z_j$ is the $j$-th position of a transformed random walk $\{Z_j\}_{j\in\mathbb{N}}$, $j^* = \max\{j \in \mathbb{N}|-\epsilon/a_i \leq Z_j \leq \epsilon/a_i\}$ is the random variable indicating the maximal position when the random walk is within the strip of boundary $\pm\epsilon/a_i$, and the last equality is due to the definition of expectation.

Under the MET algorithm setting where $a_i = 1/\mu$ and $\epsilon \gg \sigma/\mu$, we can show that the increments of the transformed random walk $\{Z_j\}$ have zero mean and variance $\sigma^2$, and many steps are necessary to reach the random walk boundary. With the Central Limit Theorem, we can assume the $Z_j$ follows normal distribution with mean $\mu_{zj} = 0$ and variance $\sigma_{zj}^2 = j\sigma^2$, and thus $|Z_j|$ follows the folded normal distribution with expectation $\mathbb{E}(|Z_j|) = \sqrt{2/\pi}\sigma\sqrt{j}$. Thus Eq. (1) can be written as

$$\frac{1}{\mu}\sum_{n=1}^{\infty}\mathbb{E}\left[\sum_{j=0}^{n-1}|Z_j|\right]\Pr(j^*=n) < \frac{1}{\mu}\sum_{n=1}^{\infty}\sum_{j=0}^{n-1}\mathbb{E}[|Z_j|]\Pr(j^*=n) = \frac{\sigma}{\mu}\sqrt{\frac{2}{\pi}}\sum_{n=1}^{\infty}\sum_{j=0}^{n-1}\sqrt{j}\,\Pr(j^*=n).$$

Using $\mathbb{E}[j^*] = \frac{\mu^2}{\sigma^2}\epsilon^2$ and $Var[j^*] = \frac{2}{3}\frac{\mu^4}{\sigma^4}\epsilon^4$ as derived in [10], we get $\mathbb{E}[(j^*)^2] = \frac{5}{3}\frac{\mu^4}{\sigma^4}\epsilon^4$. With the inequality $\sum_{j=0}^{n-1}\sqrt{j} < \frac{2}{3}n\sqrt{n}$ and $\mathbb{E}[X^{\frac{3}{4}}] \leq (\mathbb{E}[X])^{\frac{3}{4}}$, we get the upper bound:

$$\mathbb{E}[SegErr_i] < \frac{2}{3}\sqrt{\frac{2}{\pi}}\frac{\sigma}{\mu}\mathbb{E}[(j^*)^{\frac{3}{2}}] \leq \frac{2}{3}\sqrt{\frac{2}{\pi}}\frac{\sigma}{\mu}\left(\mathbb{E}[(j^*)^2]\right)^{\frac{3}{4}} = \frac{2}{3}\sqrt{\frac{2}{\pi}}(\frac{5}{3})^{\frac{3}{4}}(\frac{\mu}{\sigma})^2\epsilon^3.$$

For the lower bound, applying the triangle inequality into Eq. (1), we can get $\mathbb{E}[SegErr_i] > \frac{1}{\mu}\sum_{n=1}^{\infty}\mathbb{E}[|Z|]\Pr(j^*=n)$, where $Z = \sum_{j=0}^{n-1}Z_j$, and $Z$ follows the normal distribution since $Z_j \sim N(0, \sigma_{zj}^2)$. We can prove that $|Z|$ follows the folded normal distribution whose expectation $\mathbb{E}[|Z|] > \sigma(n-1)/\sqrt{\pi}$. Thus the lower bound is:

$$\mathbb{E}[SegErr_i] > \frac{\sigma}{\mu}\sqrt{\frac{1}{\pi}}\sum_{n=1}^{\infty}(n-1)\Pr(j^*=n) = \frac{\sigma}{\mu}\sqrt{\frac{1}{\pi}}\mathbb{E}[j^*-1] = \sqrt{\frac{1}{\pi}}(\frac{\mu}{\sigma}\epsilon^2 - \frac{\sigma}{\mu}).$$

Since $\epsilon \gg \frac{\sigma}{\mu}$, we can omit the right term $\sqrt{1/\pi} \cdot \sigma/\mu$ and finish the proof. Although the derivations are based on the MET algorithm whose slope is the reciprocal of $\mu$, we found that the mathematical forms among $\epsilon$, $\mu/\sigma$ and $SegErr_i$ are still applicable to other $\epsilon$-bounded methods, and further prove that the learned segment slopes of other methods are close to the reciprocal of expected key intervals in Appendix B. For the another independence assumption adopted by the MET algorithm, the authors discussed that the Central Limit Theorem holds for non-i.i.d. variables and the theorems can be extended accordingly [10]. We empirically show that the proposed framework is robust to these assumptions and works well for several SOTA methods on the real-world datasets (Section 4.2).

### 3.4 $\epsilon$-Learner

Now given an $\epsilon$, we have obtained the closed-form bounds of the $SegErr$ in Theorem 1, and both the upper and lower bounds are in the form of $w_1(\frac{\mu}{\sigma})^{w_2}\epsilon^{w_3}$, where $w_{1,2,3}$ are some coefficients. As the concrete values of these coefficients can be different for different datasets and different methods, we propose to learn the following trainable estimator to make the error prediction preciser:

$$SegErr = f(\epsilon, \mu, \sigma) = w_1(\frac{\mu}{\sigma})^{w_2}\epsilon^{w_3},$$

$$s.t. \sqrt{\frac{1}{\pi}} \leq w_1 \leq \frac{2}{3}\sqrt{\frac{2}{\pi}}(\frac{5}{3})^{\frac{3}{4}}, \quad 1 \leq w_2 \leq 2, \quad 2 \leq w_3 \leq 3. \tag{2}$$

With this learnable estimator, we feed data characteristic $\mu/\sigma$ of the look-ahead data and the transformed $\widetilde{SegErr}$ into it and find a suitable $\epsilon^*$ as $\left(\widetilde{SegErr}/w_1(\frac{\mu}{\sigma})^{w_2}\right)^{1/w_3}$. We will discuss the look-ahead data and the transformed $\widetilde{SegErr}$ in the following paragraphs. Now let's discuss the reasons for how this adjustment can achieve better index performance. Actually, the $\epsilon$-learner proactively plans the allocations of the total prediction error indicated by user (*i.e.*, $\tilde{\epsilon} \cdot |\mathcal{D}|$) and calculates the tolerated $\widetilde{SegErr}$ for the next segment. By adjusting current $\epsilon$ to $\epsilon^*$, the following learned segment can fully utilize the distribution information of the data and achieve better performance in terms of *N-MAE* trade-off. To be specific, when $\mu/\sigma$ is large, the local data has clear linearity, and thus we can adjust $\epsilon$ to a relatively small value to gain precise predictions; although the number of data points covered by this segment may decrease and then the number of total segments increases, such cost paid in terms of space is not larger than the benefit we gain in terms of precise predictions. Similarly, when $\mu/\sigma$ is small, $\epsilon$ should be adjusted to a relatively large value to lower the learning difficulty and absorb some non-linear data localities; in this case, we gain in terms of space while paying some costs in terms of prediction accuracy. The segment-wise adjustment of $\epsilon$ improves the overall index performance by continually and data-dependently balancing the cost of space and preciseness.

**Look-ahead Data.** To make the training and inference of the $\epsilon$-learner light-weight, we propose to look ahead a few data $\mathcal{D}'$ to reflect the characteristics of the following data localities. Specifically, we leverage a small subset $\mathcal{D}' \subset \mathcal{D} \setminus \bigcup_{j<i} \mathcal{D}_j$ to estimate the value $\mu/\sigma$ for the following data. In practice, we set the size of $\mathcal{D}'$ to be 404 when learning the first segment as initialization, and $\left(\frac{1}{(i-1)}\sum_{j=1}^{i-1} Len(\mathcal{D}_j)\right) \cdot \rho$ for the other following segments. Here $\rho$ is a pre-defined parameter indicating the percentage that is relative to the average number of covered keys for learned segments, considering that the distribution of $\mu/\sigma$ can be quite different to various datasets. As for the first segment, according to the literature [16], the sample size 404 can provide a 90% confidence intervals for a coefficient of variance $\sigma/\mu \leq 0.2$.

$\widetilde{SegErr}$ **and Optimization.** As aforementioned, taking the user-expected $\tilde{\epsilon}$ as input, we aim to reflect the impact of $\tilde{\epsilon}$ with a transformed proxy quantity $\widetilde{SegErr}$ such that the $\epsilon$-learner can choose suitable $\epsilon^*$ to meet users' preference while achieving better *N-MAE* trade-off. Specifically, we make the value of $\widetilde{SegErr}$ updatable, and update it to be $\widetilde{SegErr} = w_1(\hat{\mu}/\hat{\sigma})^{w_2}\tilde{\epsilon}^{w_3}$ once a new segment is learned, where $\hat{\mu}/\hat{\sigma}$ is the mean value of all the processed data so far. This strategy enables us to promptly incorporate both the user preference and the data distribution into the calculation of $\widetilde{SegErr}$. As for the optimization of the light-weight model, *i.e.*, $f(\epsilon, \mu, \sigma)$ that contains only three learnable parameters $w_{1,2,3}$, we adopt the projected gradient descent [4, 8] with the parameter constraints in Eq. (2). In this way, we only need to track a few statistics and learn the $\epsilon$ estimator in an efficient one-pass manner. The overall algorithm is summarized in Appendix D.

## 4 Experiments

### 4.1 Experimental Settings

**Baselines.** We apply our framework into several SOTA $\epsilon$-bounded learned index methods that use different mechanisms to determine the parameters of segments $\{S_i\}$. Among them, *MET* [10] fixes the segment slope as the reciprocal of the expected key interval. *FITing-Tree* [14] and *Radix-Spline* [18] adopt a greedy shrinking cone algorithm and a spline interpolating algorithm respectively. *PGM* [12] adopts a convex hull based algorithm to achieve the minimum number of learned segments. More introduction and implementation details are in Appendix F.

**Datasets.** We use several widely adopted datasets that differ in data scales and distributions [19, 14, 9, 12, 21]. *Weblogs* and *IoT* contain 715M log entries from a university web server and 26M event entries from different IoT sensors respectively, in which the keys are log timestamps. *Map* dataset contains location coordinates around the world [25], and the keys are longitudes of 200M places. *Lognormal* is a synthetic dataset whose key intervals follow the lognormal distribution. We generate 20M keys with 40 partitions having different generation parameters to simulate the varied data characteristics among different localities. More details and visualization are in Appendix G.

**Evaluation Metrics.** We evaluate the index performance in terms of its size, prediction preciseness, and the total querying time. Specifically, we report the number of learned segments $N$, the index size in bytes, the *MAE* as $\frac{1}{|D|} \sum_{(x,y) \in D} |y - \mathbf{S}(x)|$, and the total querying time per query in ns (*i.e.*, we perform querying operations for all the indexed data, record the total time of getting the payloads given the keys, and report the time that is averaged over all the queries). For a quantitative comparison w.r.t. the trade-off improvements, we calculate the **A**rea **U**nder the $N$-$MAE$ **C**urve (AUNEC) where the x-axis and y-axis indicate $N$ and *MAE* respectively. For AUNEC metric, the smaller, the better.

### 4.2 Overall Index Performance

$N$-*MAE* **Trade-off Improvements.** In Table 1, we summarize the AUNEC improvements in percentage brought by the proposed framework of all the baseline methods on all the datasets. We also illustrate the $N$-*MAE* trade-off curves for some cases in Figure 2, where the blue curves indicate the results achieved by fixed $\epsilon$ version while the red curves are for dynamic $\epsilon$. Other baselines and datasets yield similar curves, which we include in Appendix H due to the space limitation. These results show that the dynamic $\epsilon$ versions of all the baseline methods achieve much better $N$-*MAE* trade-off ($-15.66\%$ to $-22.61\%$ averaged improvements as smaller AUNEC indicates better performance), demonstrating the effectiveness and the wide applicability of the proposed framework. As discussed in previous sections, datasets usually have diverse key distributions at different data localities, and the proposed framework can data-dependently adjust $\epsilon$ to fully utilize the distribution information of data localities and thus achieve better index performance in terms of $N$-*MAE* trade-off. Here the Map dataset has significant non-linearity caused by spatial characteristics, and it is hard to fit using linear segments (all baseline methods learn linear segments), thus relatively small improvements are achieved.

Table 1: The AUNEC relative *improvements* for learned index methods with dynamic $\epsilon$.

|  | Weblogs | IoT | Map | Lognormal | Average |
|---|---|---|---|---|---|
| MET | -25.87% | -7.66% | -7.63% | -21.48% | -15.66% |
| FITing-Tree | -31.18% | -25.56% | -4.94% | -28.24% | -22.48% |
| Radix-Spline | -28.37% | -24.59% | -6.14% | -31.32% | -22.61% |
| PGM | -22.42% | -25.01% | -7.18% | -6.52% | -15.28% |

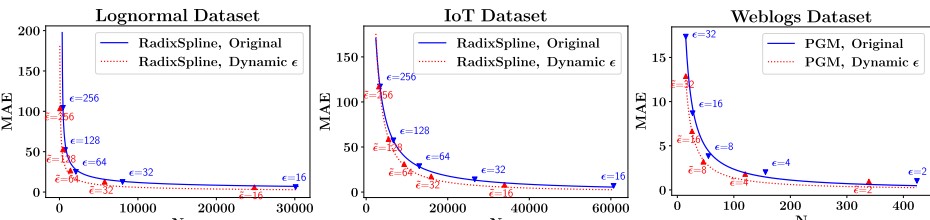

Figure 2: The $N$-*MAE* trade-off curves for learned index methods.

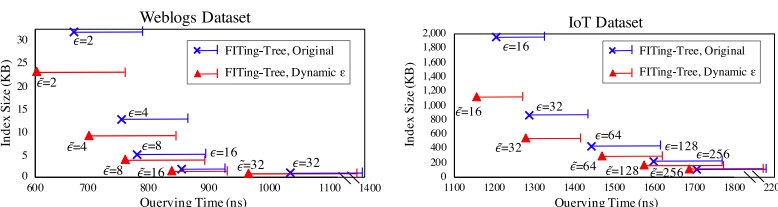

Figure 3: Improvements in terms of querying time for learned index methods with dynamic $\epsilon$.

**Querying Time Improvements.** Recall that the querying time of each data point is in $O(\log(N) + \log(|y - \hat{y}|))$ as we mentioned in Section 3.1, where $N$ and $|y - \hat{y}|$ are inversely impacted by $\epsilon$. To examine whether the performance improvements w.r.t. $N$-*MAE* trade-off (*i.e.*, Table 1) can lead

to better querying efficiency in real-world systems, we show the averaged total querying time per query and the actual learned index size in bytes for two scenarios in Figure 3. We also mark the 99th percentile (P99) latency as the right bar. We can observe that the dynamic $\epsilon$ versions indeed gain faster average querying speed, since we improve both the term $N$ as well as the term $|y - \hat{y}|$ via adaptive adjustment of $\epsilon$. Besides, we find that the dynamic version achieves comparable or even better P99 results than the static version, due to the fact that our method effectively adjust $\epsilon$ based on the expected $\tilde{\epsilon}$ and data characteristic, making the $\{\epsilon_i\}$ fluctuated within a moderate range and leading to a good robustness. The similar conclusion can be drawn from other baselines and datasets, and we present their results in Appendix H. Another thing to note is that, this experiment also verifies the usability of our framework in which users can flexibly set the expected $\tilde{\epsilon}$ to meet various space-time preferences just as they set $\epsilon$ in the original learned index methods.

**Index Building Cost.** Comparing with the original learned index methods that adopt a fixed $\epsilon$, we introduces extra computation to dynamically adjust $\epsilon$ in the index building stage. Does this affect the efficiency of original methods? Here we report the relative increments of building times in Table 2. From it, we can observe that the proposed dynamic $\epsilon$ framework achieves comparable building times to all the original learned index methods on all the datasets, showing the efficiency of our framework since it retains the online learning manner with the same complexity as the original methods (both in $O(|\mathcal{D}|)$). Note that we only need to pay this extra cost once, *i.e.*, building the index once, and then the index structures can accelerate the frequent data querying operations for real-world applications.

Table 2: Building time *increments* in percentage for learned index methods with dynamic $\epsilon$.

|  | Weblogs | IoT | Map | Lognormal | Average |
|---|---|---|---|---|---|
| MET | 10.54% | 5.14% | 8.33% | 5.26% | 7.32% |
| FITing-Tree | 10.70% | 1.88% | 5.35% | 5.23% | 5.79% |
| Radix-Spline | 10.19% | 1.64% | 3.85% | 8.96% | 6.16% |
| PGM | 16.76% | 2.20% | 1.28% | 21.29% | 10.38% |

### 4.3 Ablation Study of Dynamic $\epsilon$

To gain further insights about how the proposed dynamic $\epsilon$ framework works, we compare the proposed one with three dynamic $\epsilon$ variants: (1) *Random $\epsilon$* is a vanilla version that randomly choose $\epsilon$ from $[0, 2\tilde{\epsilon}]$ when learning each new segment; (2) *Polynomial Learner* differs our framework with another polynomial function $SegErr(\epsilon) = \theta_1 \epsilon^{\theta_2}$ where $\theta_1$ and $\theta_2$ are trainable parameters; (3) *Least Square Learner* differs our framework with an optimal (but very costly) strategy to learn $f(\epsilon, \mu, \sigma)$ with the least square regression.

Table 3: The AUNEC relative changes of dynamic $\epsilon$ variants compared to the proposed framework.

|  | Weblogs | IoT | Map | Lognormal | Average |
|---|---|---|---|---|---|
| Random $\epsilon$ | +70.94% | +68.19% | +53.29% | +73.38% | +66.45% |
| Polynomial Learner | +49.32% | +40.57% | +7.71% | +42.77% | +35.09% |
| Least Square Learner | +4.44% | +9.32% | +2.04% | −17.63% | −0.46% |

We summarize the AUNEC changes in percentage compared to the proposed framework in Table 3. Here we only report the results for FITing-Tree due to the space limitation and similar results can be observed for other methods. Recall that for AUNEC, the smaller, the better. From this table, we have the following observations: (1) The *Random $\epsilon$* version achieves much worse results than the proposed dynamic $\epsilon$ framework, showing the necessity and effectiveness of learning the impact of $\epsilon$. (2) The *Polynomial Learner* achieves better results than the *Random $\epsilon$* version while still have a large performance gap compared to our proposed framework. This indicates the usefulness of the derived theoretical results that link the index performance, the $\epsilon$ and the data characteristics together. (3) For the *Least Square Learner*, we can see that it achieves similar AUNEC results compared with the proposed framework. However, it has higher computational complexity and pays the cost of much larger building times, *e.g.*, $14\times$ and $53\times$ longer building times on IoT and Map respectively. These results demonstrate the effectiveness and efficiency of the proposed framework that adjusts $\epsilon$ based on the theoretical results, which will be validated next.

### 4.4 Theoretical Results Validation

We study the impact of $\epsilon$ on $SegErr_i$ for the MET algorithm in Theorem 1, where the derivations are based on the setting of the slope condition $a_i = 1/\mu$. To confirm that the proposed framework

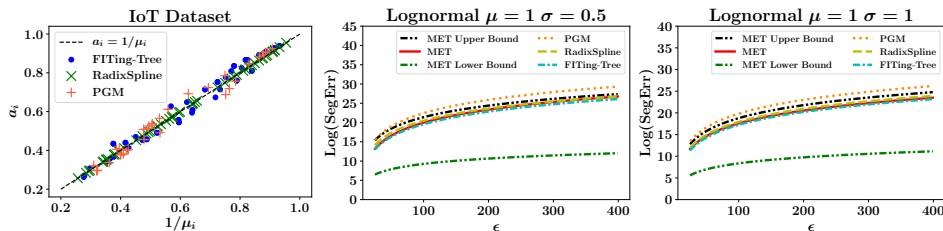

Figure 4: Learned slopes.    Figure 5: Illustration of the derived bounds.

also works well with other $\epsilon$-bounded learned index methods, we analyze the learned slopes of other
$\epsilon$-bounded methods in Appendix B. In summary, we prove that for a segment $S_i : y = a_i x + b_i$
whose covered data is $\mathcal{D}_i$ and the expected key interval of $\mathcal{D}_i$ is $\mu_i$, then $a_i$ concentrates on $1/\mu_i$
within $2\epsilon/(\mathbb{E}[Len(\mathcal{D}_i)] - 1)$ relative deviations. Here we plot the learned slopes of baseline learned
index methods in Figure 4. We can see that the learned slopes of other methods indeed center along
the line $a_i = 1/\mu_i$, showing the close connections among these methods and confirming that the
proposed framework can work well with other $\epsilon$-bounded learned index methods.

We further compare the theoretical bounds with the actual $SegErr_i$ for all the adopted learned
index methods. In Figure 5, we only show the results on Lognormal dataset due to space limitation.
As expected, we can see that the MET method has the actual $SegErr_i$ within the derived bounds,
verifying the correctness of the Theorem 1. Besides, the other $\epsilon$-bounded methods show the same
trends with the MET method, providing the evidence that these methods have the same mathematical
forms as we derived, and thus the $\epsilon$-learner also works well with them.

### 4.5  Case Study

We visualize the partial learned segments for FITing-Tree with
fixed and dynamic $\epsilon$ on IoT dataset in Figure 6, where the $N$ and
$\sum SegErr_i$ indicates the number of learned segments and the
total prediction error for the shown segments respectively. The
$\overrightarrow{\mu/\sigma}$ indicates the characteristics of covered data $\{\mathcal{D}_i\}$. We can
see that our dynamic framework helps the learned index gain
both smaller space (7 v.s. 4) and smaller total prediction errors
(48017 v.s. 29854). Note that $\epsilon$s within $\overrightarrow{\epsilon_i}$ are diverse due to the
diverse linearity of different data localities: For the data whose
positions are within about $[30000, 30600]$ and $[34700, 35000]$,
the proposed framework chooses large $\epsilon$s as their $\mu/\sigma$s are small,
and by doing so, it achieves smaller $N$ than the fixed version by
absorbing these non-linear localities; For the data at the middle
part, they have clear linearity with large $\mu/\sigma$s, and thus the
proposed framework adjusts $\epsilon$ as 19 and 10 that are smaller than
32 to achieve better precision. These experimental observations
are consistent with our analysis in the paragraph under Eq. (2),
and clearly confirm that the proposed framework adaptively adjusts $\epsilon$ based on data characteristics.

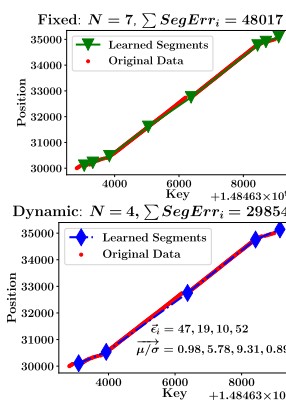

Figure 6: Visualization of the learned index (partial) on IoT for FITing-Tree with fixed $\epsilon = 32$ and dynamic version ($\tilde{\epsilon} = 32$).

## 5  Conclusions

Existing learned index methods introduce an important hyper-parameter $\epsilon$ to provide a worst-case
preciseness guarantee and meet various space-time user preferences. In this paper, we provide
formal analysis about the relationships among $\epsilon$, data local characteristics and the introduced quantity
$SegErr_i$ for each learned segment, which is the product of the number of covered keys and *MAE*,
and thus embeds the space-time trade-off. Based on the derived bounds, we present a pluggable
dynamic $\epsilon$ framework that leverages an $\epsilon$-learner to data-dependently adjust $\epsilon$ and achieve better
index performance in terms of space-time trade-off. A series of experiments verify the effectiveness,
efficiency and usability of the proposed framework.

We believe that our work contributes a deeper understanding of how the $\epsilon$ impacts the index perfor-
mance, and enlightens the exploration of fine-grained trade-off adjustments by considering data local
characteristics. Our study also opens several interesting future works. For example, we can apply the
proposed framework to other problems in which the piece-wise approximation algorithms with fixed
$\epsilon$ are used while still requiring space-time trade-off, such as similarity search and lossy compression
for time series data [5, 33, 3, 26].

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
