**Appendices for the Submission: Learned Index with Dynamic $\epsilon$**

522 Our appendices include the following content:

- 523 • Sec.A: the full **proof** of Theorem 1.

- 524 • Sec.B: the analysis about the **learned slopes** of other $\epsilon$-bounded methods.

- 525 • Sec.C: the details of the binary search and exponential search, and the connections between
  526 **prediction error** and these specific **searching strategies**.

- 527 • Sec.D: the summarized **algorithm** of the proposed method.

- 528 • Sec.E: the discussion about how the proposed framework **inherits the good abilities** of
  529 existing learned index methods.

- 530 • Sec.F: the **implementation details** of experiments.

- 531 • Sec.G: the detailed introduction and visualization of the adopted **datasets**.

- 532 • Sec.H: more **experimental results** including the overall index performance, ablation study
  533 and theoretical validation on other datasets and methods. Besides, we explore an indica-
  534 tive quantity (the $CV$ value) to provide further insight into the rational of the proposed
  535 framework.

## A Proof of Theorem 1

537 Given a learned segment $S_i : y = a_i x + b_i$, denote $c_i$ as the stored position of the last covered data
538 for the $(i-1)$-th segment ($c_1 = 0$ for the first segment). We can write the expectation of $SegErr_i$
539 for the segment $S_i$ as the following form:

$$\mathbb{E}[SegErr_i] = \mathbb{E}\left[ \sum_{j=0}^{(j^*-1)} |a_i X_j + b_i - (j + c_i + 1)| \right],$$

540 where $j^*$ indicates the length of the segment, and $X_j$ indicates the $j$-th key covered by the segment $S_i$.
541 As studied in [10], the linear-approximation problem with $\epsilon$ guarantee can be modeled as random walk
542 processes. Specifically, $X_j = X_0 + \sum_{k=0}^{j} G_k$ (for $j \in \mathbb{Z}_{>0}$) where $G_k$ is the key increment variable
543 whose mean and variance is $\mu$ and $\sigma^2$ respectively. Denote the $Z_j = X_j - j/a_i + (b_i - c_i - 1)/a_i$ as the
544 $j$-th position of the transformed random walk $\{Z_j\}_{j \in \mathbb{N}}$, and $j^* = \max\{j \in \mathbb{N}| -\epsilon/a_i \le Z_j \le \epsilon/a_i\}$
545 as the random variable indicating the maximal position when the random walk is within the strip of
546 boundary $\pm\epsilon/a_i$. The expectation can be rewritten as

$$\mathbb{E}\left[ \sum_{j=0}^{(j^*-1)} |a_i X_j - j + (b_i - c_i - 1)| \right] = a_i \mathbb{E}\left[ \sum_{j=0}^{(j^*-1)} |Z_j| \right]$$
$$= a_i \sum_{n=1}^{\infty} \mathbb{E}\left[ \sum_{j=0}^{n-1} |Z_j| \right] \Pr(j^* = n). \tag{3}$$

547 The last equality in Eq. (3) is due to the definition of expectation. Following the MET algorithm that
548 the $S_i$ goes through the point $(X_0, Y_0 = c_i + 1)$, we get $b_i = -a_i X_0 + c_i + 1$ and we can rewrite
549 $Z_j$ as the following form:

$$Z_0 = 0, \quad Z_j \overset{j \ge 0}{=} X_j - X_0 - j/a_i = \sum_{k=1}^{j} G_k - j/a_i$$
$$= \sum_{k=1}^{j} (G_k - 1/a_i) = \sum_{k=1}^{j} (W_k),$$

550 where $W_k$ is the walk increment variable of $Z_j$, $\mathbb{E}[W_k] = \mu - 1/a_i$ and $Var[W_k] = \sigma^2$. Under
551 the MET algorithm setting where $a_i = 1/\mu$ and $\varepsilon \gg \sigma/\mu$, the transformed random walk $\{Z_j\}$ has
552 increments with zero mean and variance $\sigma^2$, and many steps are necessary to reach the random walk

boundary. With the Central Limit Theorem, we can assume that $Z_j$ follows the normal distribution with mean $\mu_{zj}$ and variance $\sigma_{zj}^2$, and thus $|Z_j|$ follows the folded normal distribution:

$$Z_j \sim \mathcal{N}\big((\mu - 1/a_i)j, j\sigma^2\big),$$

$$\mathbb{E}(|Z_j|) = \mu_{zj}[1 - 2\Phi(-\mu_{zj}/\sigma_{zj})] + \sigma_{zj}\sqrt{2/\pi}\exp(-\mu_{zj}^2/2\sigma_{zj}^2),$$

where $\Phi$ is the normal cumulative distribution function. For the MET algorithm, $a_i = 1/\mu$ and thus the $\mu_{zj} = 0$, $\sigma_{zj} = \sigma\sqrt{j}$, and $\mathbb{E}(|Z_j|) = \sqrt{2/\pi}\sigma\sqrt{j}$. Then the Eq. (3) can be written as

$$\frac{1}{\mu}\sum_{n=1}^{\infty}\mathbb{E}\left[\sum_{j=0}^{n-1}|Z_j|\right]\Pr(j^* = n) < \frac{1}{\mu}\sum_{n=1}^{\infty}\sum_{j=0}^{n-1}\mathbb{E}\left[|Z_j|\right]\Pr(j^* = n)$$

$$= \frac{\sigma}{\mu}\sqrt{\frac{2}{\pi}}\sum_{n=1}^{\infty}\sum_{j=0}^{n-1}\sqrt{j}\Pr(j^* = n). \tag{4}$$

For the inner sum term in Eq. (4), we have $(\sum_{j=0}^{n-1}\sqrt{j}) < \frac{2}{3}n\sqrt{n}$ since

$$\sum_{j=0}^{n-1}\sqrt{j} < \sum_{j=0}^{n-1}\sqrt{j} + \frac{\sqrt{n}}{2} < \int_0^n \sqrt{x}\,dx = \frac{2}{3}n\sqrt{n},$$

then the result in Eq. (4) becomes

$$\mathbb{E}[SegErr_i] < \frac{2}{3}\sqrt{\frac{2}{\pi}}\frac{\sigma}{\mu}\sum_{n=1}^{\infty}n\sqrt{n}\Pr(j^* = n)$$

$$= \frac{2}{3}\sqrt{\frac{2}{\pi}}\frac{\sigma}{\mu}\mathbb{E}[(j^*)^{\frac{3}{2}}] = \frac{2}{3}\sqrt{\frac{2}{\pi}}\frac{\sigma}{\mu}\mathbb{E}\left[((j^*)^2)^{\frac{3}{4}}\right]$$

$$\leq \frac{2}{3}\sqrt{\frac{2}{\pi}}\frac{\sigma}{\mu}\left(\mathbb{E}[(j^*)^2]\right)^{\frac{3}{4}},$$

where the last inequality holds due to the Jensen inequality $\mathbb{E}[X^{\frac{3}{4}}] \leq (\mathbb{E}[X])^{\frac{3}{4}}$. Using $\mathbb{E}[j^*] = \frac{\mu^2}{\sigma^2}\epsilon^2$ and $Var[j^*] = \frac{2}{3}\frac{\mu^4}{\sigma^4}\epsilon^4$ derived in MET algorithm [10], we get $\mathbb{E}[(j^*)^2] = \frac{5}{3}\frac{\mu^4}{\sigma^4}\epsilon^4$, which yields the following upper bound:

$$\mathbb{E}[SegErr_i] < \frac{2}{3}\sqrt{\frac{2}{\pi}}(\frac{5}{3})^{\frac{3}{4}}(\frac{\mu}{\sigma})^2\epsilon^3.$$

For the lower bound, applying the triangle inequality into the Eq. (3), we have

$$\frac{1}{\mu}\sum_{n=1}^{\infty}\mathbb{E}\left[\sum_{j=0}^{n-1}|Z_j|\right]\Pr(j^* = n)$$

$$> \frac{1}{\mu}\sum_{n=1}^{\infty}\mathbb{E}\left[|\sum_{j=0}^{n-1}Z_j|\right]\Pr(j^* = n) \tag{5}$$

$$= \frac{1}{\mu}\sum_{n=1}^{\infty}\mathbb{E}\left[|Z|\right]\Pr(j^* = n),$$

where $Z = \sum_{j=0}^{n-1}Z_j$. Since $Z_j \sim N(0, \sigma_{zj}^2)$, the $Z$ follows the normal distribution:

$$Z \sim \mathbb{N}\Big(\mu_Z = 0,\ \ \sigma_Z^2 = \sum_{j=0}^{n-1}\sigma_{zj}^2 + \sum_{j=0}^{n-1}\sum_{k=0, k\neq j}^{n-1}r_{jk}\sigma_{zj}\sigma_{zk}\Big),$$

where $r_{jk}$ is the correlation between $Z_j$ and $Z_k$. Since $\mu_Z = 0$, the $|Z|$ follows the folded normal distribution with $\mathbb{E}[|Z|] = \sigma_Z\sqrt{2/\pi}$. Since the random walk $\{Z_j\}$ is a process with i.i.d. increments, the correlation $r_{jk} \geq 0$. With $\sigma_{zj} = \sigma\sqrt{j} > 0$ and $r_{jk} \geq 0$, we have

$$\mathbb{E}[|Z|] > \sqrt{\frac{2}{\pi}}\sum_{j=0}^{n-1}\sigma_{zj} > \sigma\sqrt{n(n-1)/\pi} > \frac{\sigma(n-1)}{\sqrt{\pi}},$$

and the result in Eq. (5) becomes:

$$\mathbb{E}[SegErr_i] > \frac{1}{\mu} \sum_{n=1}^{\infty} \mathbb{E}\left[ |\sum_{j=0}^{n-1} Z_j| \right] \Pr(j^* = n)$$

$$> \frac{\sigma}{\mu} \sqrt{\frac{1}{\pi}} \sum_{n=1}^{\infty} (n-1) \Pr(j^* = n)$$

$$= \frac{\sigma}{\mu} \sqrt{\frac{1}{\pi}} \mathbb{E}[j^* - 1] = \sqrt{\frac{1}{\pi}} (\frac{\mu}{\sigma} \epsilon^2 - \frac{\sigma}{\mu}).$$

Since $\epsilon \gg \frac{\sigma}{\mu}$, we can omit the right term $\sqrt{\frac{1}{\pi}} \frac{\sigma}{\mu}$ and finish the proof.

## B  Learned Slopes of Other $\epsilon$-Bounded Methods

As shown in Theorem 1, we have known how $\epsilon$ impacts the $SegErr_i$ of each segment learned by the MET algorithm, where the theoretical derivations largely rely on the slope condition $a_i = 1/\mu$. Here we prove that for other $\epsilon$-bounded methods, the learned slope of each segment (*i.e.*, $a_i$ of $S_i$) concentrates on the reciprocal of the expected key interval as shown in the following Theorem.

**Theorem 2.** *Given an $\epsilon \in \mathbb{Z}_{>1}$ and an $\epsilon$-bounded learned index algorithm $\mathcal{A}$. For a linear segment $S_i : y = a_i x + b_i$ learned by $\mathcal{A}$, denote its covered data and the number of covered keys as $\mathcal{D}_i$ and $Len(\mathcal{D}_i)$ respectively. Assuming the expected key interval of $\mathcal{D}_i$ is $\mu_i$, the learned slope $a_i$ concentrates on $\tilde{a} = 1/\mu_i$ with bounded relative difference:*

$$(1 - \frac{2\epsilon}{\mathbb{E}[Len(\mathcal{D}_i)] - 1})\tilde{a} \quad \le \quad E[a_i] \quad \le \quad (1 + \frac{2\epsilon}{\mathbb{E}[Len(\mathcal{D}_i)] - 1})\tilde{a}.$$

*Proof.* For the learned linear segment $S_i$, denote its first predicted position and last predicted position as $y_0'$ and $y_n'$ respectively, we have its slope $a_i = \frac{y_n' - y_0'}{x_n - x_0}$. Notice that $y_0 - \epsilon \le y_0' \le y_0 + \epsilon$ and $y_n - \epsilon \le y_n' \le y_n + \epsilon$ due to the $\epsilon$ guarantee, we have $y_n - y_0 - 2\epsilon \le y_n' - y_0' \le y_n - y_0 + 2\epsilon$ and the expectation of $a_i$ can be written as

$$\mathbb{E}[\frac{y_n - y_0 + 2\epsilon}{x_n - x_0}] \quad \le \quad E[a_i] = \frac{y_n' - y_0'}{x_n - x_0} \quad \le \quad \mathbb{E}[\frac{y_n - y_0 + 2\epsilon}{x_n - x_0}].$$

Note that for any learned segment $S_i$ whose first covered data is $(x_0, y_0)$ and last covered data is $(x_n, y_n)$, we have $\mathbb{E}[\frac{x_n - x_0}{y_n - y_0}] = \mu_i$ and thus the inequalities become

$$\frac{1}{\mu} - \mathbb{E}[\frac{2\epsilon}{x_n - x_0}] \quad \le \quad E[a_i] \quad \le \quad \frac{1}{\mu} + \mathbb{E}[\frac{2\epsilon}{x_n - x_0}].$$

Since $\tilde{a} = 1/\mu_i$ and $\mathbb{E}[x_n - x_0] = (\mathbb{E}[Len(\mathcal{D}_i)] - 1)\mu_i$, we finish the proof. $\square$

The Theorem 2 shows that the relative deviations between learned slope $a_i$ and $\tilde{a}$ are within $2\epsilon/(\mathbb{E}[Len(\mathcal{D}_i)] - 1)$. For the MET and PGM learned index methods, we have the following corollary that depicts preciser deviations without the expectation term $\mathbb{E}[Len(\mathcal{D}_i)]$.

**Corollary 2.1.** *For the MET method [10] and the optimal $\epsilon$-bounded linear approximation method that learns the largest segment length used in PGM [12], the slope relative differences are at $O(1/\epsilon)$.*

*Proof.* We note that the segment length of a learned segment is at $O(\epsilon^2)$ for the MET algorithm, which is proved in the Theorem 1 of [10]. Since PGM achieves the largest learned segment length that is larger than the one of the MET algorithm, we finish the proof. $\square$

## C  Connecting Prediction Error with Searching Strategy

As we mentioned in Section 3.1, we can find the true position of the queried data point in $O(\log(N) + \log(|\hat{y} - y|))$ where $N$ is the number of learned segments and $|\hat{y} - y|$ is the absolute prediction error. A binary search or exponential search finds the stored true position $y$ based on $\hat{y}$. It is worth pointing

out that the searching cost in terms of searching range $|\hat{y} - y|$ of binary search strategy corresponds to the maximum absolute prediction error $\epsilon$, whereas the one of exponential search corresponds to the mean absolute prediction error (*MAE*). In this paper, we decouple the quantity $SegErr_i$ as the product of $Len(\mathcal{D}_i)$ and $MAE(\mathcal{D}_i|S_i)$ in the derivation of Theorem 1. Built upon the theoretical analysis, we adopt exponential search in experiments to better leverage the predictive models.

To clarify, let's consider a learned segment $S_i$ with its covered data $\mathcal{D}_i$. Let $|\hat{y_k} - y_k|$ be the absolute prediction error of $k$-th data point covered by this segment, and $\epsilon_i$ be the maximum absolute prediction error of $S_i$, *i.e.*, $|\hat{y_k} - y_k| \leq \epsilon_i$ for all $k \in [len(\mathcal{D}_i)]$.

- The binary search is conducted within the searching range $[\hat{y_k} \pm \epsilon_i]$ for each data point [2], thus the mean search range is $\frac{1}{len(\mathcal{D}_i)} \sum_{k=1}^{len(\mathcal{D}_i)} 2\epsilon_i = O(\epsilon_i)$, which is independent of the preciseness of the learned segment and an upper bound of $MAE(\mathcal{D}_i|S_i)$.

- The exponential search first finds the searching range where the queried data may exist by centering around the $\hat{y}$, repeatedly doubling the range $[\hat{y} \pm 2^q]$ where the integer $q$ grows from 0, and comparing the queried data with the data points at positions $\hat{y} \pm 2^q$. After finding the specific range such that a $q_k$ satisfies $2^{\log(q_k)-1} \leq |\hat{y_k} - y_k| \leq 2^{\lceil \log(q_k) \rceil}$ for the $k$-th data, an binary search is conducted to find the exact location. In this way, the mean search range is $\frac{1}{len(\mathcal{D}_i)} \sum_{k=1}^{len(\mathcal{D}_i)} (2^{\lceil \log(q_k) \rceil + 1}) = O\big(MAE(\mathcal{D}_i|S_i)\big)$, which can be much smaller than $O(\epsilon_i)$ especially for strong predictive models and the datasets having clear linearity.

## D  The Algorithm of Dynamic $\epsilon$ Adjustment

---
**Algorithm** Dynamic $\epsilon$ Adjustment with Pluggable $\epsilon$ Learner

---
**Input:** $\mathcal{D}$: Data to be indexed, $\mathcal{A}$: Learned index algorithm, $\tilde{\epsilon}$: Expected $\epsilon$, $\rho$: Length percentage for look-ahead data
**Output: S**: Learned segments with varied $\epsilon$s
 1: initial parameters $w_{1,2,3}$ of the learned function: $f(\epsilon, \mu, \sigma) = w_1(\frac{\mu}{\sigma})^{w_2} \tilde{\epsilon}^{w_3}$
 2: initial mean length of learned segments so far: $Len(\mathcal{D}_\mathbf{S}) \leftarrow 404$
 3: $\mathbf{S} \leftarrow \varnothing, \ (\hat{\mu}/\hat{\sigma}) \leftarrow 0$
 4: **repeat**
 5:     Get data statistic:
 6:        $(\mu/\sigma) \leftarrow lookahead(\mathcal{D}, Len(\mathcal{D}_\mathbf{S}) \cdot \rho)$
 7:     Adjust $\epsilon$ based on the learner:
 8:        $\epsilon^* \leftarrow \left( \widetilde{SegErr}/w_1(\frac{\mu}{\sigma})^{w_2} \right)^{1/w_3}$
 9:     Learn new segment $S_i$ using adjusted $\epsilon^*$:
10:        $[\mathcal{S}_i, \mathcal{D}_i] \leftarrow \mathcal{A}(\mathcal{D}, \epsilon^*)$
11:        $\mathbf{S} \leftarrow \mathbf{S} \cup \mathcal{S}_i$
12:        $\mathcal{D} \leftarrow \mathcal{D} \setminus \mathcal{D}_i, \ \mathcal{D}_\mathbf{S} \leftarrow \mathcal{D}_\mathbf{S} \cup \mathcal{D}_i$
13:     Online update $Len(\mathcal{D}_\mathbf{S})$:
14:        $Len(\mathcal{D}_\mathbf{S}) \leftarrow$ running-mean$\big(Len(\mathcal{D}_\mathbf{S}), Len(\mathcal{D}_i)\big)$
15:        $(\hat{\mu}/\hat{\sigma}) \leftarrow$ running-mean$\big((\hat{\mu}/\hat{\sigma}), (\mu/\sigma)\big)$
16:     Train the learner with ground-truth:
17:        $w_{1,2,3} \leftarrow optimize(f, S_i, SegErr_i)$
18:        $\widetilde{SegErr} \leftarrow w_1(\hat{\mu}/\hat{\sigma})^{w_2} \tilde{\epsilon}^{w_3}$
19: **until** $\mathcal{D} = \varnothing$

---

In Section 3.4, we provide detailed description about the initialization and adjustment sub-procedures. The $lookahead()$ and $optimize()$ are in the "**Look-ahead Data**" and "$\widetilde{SegErr}$ **and Optimization**" paragraph respectively.

---
[2]The lower bound and upper bounds of searching ranges should be constricted to 0 and $len(\mathcal{D}_i)$ respectively. For brevity, we omit the corner cases when comparing these two searching strategies as they both need to handle the out-of-bounds scenario.

# E   Inheriting the Abilities of Existing Works

In this Section, we discuss the benefits of our proposed framework brought by its pluggable property with two example scenarios, the dynamic data update and hard limitation on user-required index size.

We note that the data insert operation has been discussed in the adopted baseline methods, FITing-Tree [14] and PGM [12]. More importantly, neither of these two methods altered the notion of $\epsilon$ when dealing with the data insertion, and they still relied on their $\epsilon$-bounded piece-wise segmentation algorithms. The proposed framework is still valid when using their respective solutions to handle the data insertion. Specifically, FITing-Tree proposes to introduce a buffer for each learned segment, which is used to store the inserted keys, and when the buffer is full, the data covered by the segment will be re-segmented (see Section 5 in [13]). PGM adopts a logarithmic method [28, 27] that maintains a series of sorted sets $\{S_0, S_1, ..., S_b\}$ where $b = \theta(\log(|\mathcal{D}|))$, and builds multiple PGM-INDEX models over the sets. When a key $x$ is inserted, a new PGM-INDEX will be built over the merged sets (see Section 3 in [12]). In general, these solutions proposed by existing methods for inserting keys are based on *re-indexing for a piece of data along with the inserted data*, and the *re-indexing processes are the same as the original piece-wise linear segmentation processes* but for different data, therefore, we can still apply the proposed dynamic-$\epsilon$ framework for these methods in insertion scenarios just like we adjust $\epsilon$ and learn index according to the new data to be re-indexed.

For the hard size limitation case, we observe that the existing work PGM introduced a multi-criteria variant that auto-tunes itself with pre-defined hard size requirement from users. Our proposed framework is pluggable and still valid when using the PGM variant to handle the size requirement. Specifically, given a space constraint, the multi-criteria PGM propose to iteratively estimate the relationship between $\epsilon$ and $size$ with a learnable function $size(\epsilon) = a\epsilon^{-b}$, and automatically outputs the index that minimizes its query time via different estimated $\epsilon$s. Given a size requirement, we can just do the same thing in dynamic $\epsilon$ scene by setting our $\tilde{\epsilon}$ as $\epsilon$ estimated by the original PGM method.

# F   Implementation Details

All the experiments are conducted on a Linux server with an Intel Xeon Platinum 8163 2.50GHz CPU. We first introduce more details and the implementation of baseline learned index methods. *MET* [10] fixes the segment slope as the reciprocal of the expected key interval, and goes through the first available data point for each segment. *FITing-Tree* [14] adopts a greedy shrinking cone algorithm and the learned segments are organized with a B$^+$-tree. Here we use the stx::btree (v0.9) implementation [2] and set the filling factors of inner nodes and leaf nodes as $100\%$, *i.e.*, we adopt the full-paged filling manner. *Radix-Spline* [18] adopts a greedy spline interpolating algorithm to learn spline points, and the learned spline segments are organized with a flat radix table. We set the number of radix bits as $r = 16$ for the Radix-Spline method, which means that the leveraged radix table contains $2^{16}$ entries. *PGM* [12] adopts a convex hull based algorithm to achieve the minimum number of learned segments, and the segments can be organized with the help of binary search, CSS-Tree [29] and recursive structure. Here we implement the recursive version since it beats the other two variants in terms of indexing performance. For all the baselines and our method, we adopt exponential search to better leverage the predictive models since the query complexity using exponential search corresponds the preciseness of models (*MAE*) as we analyzed in Appendix C.

We then describe a few additional details of the proposed framework in terms of the $\epsilon$-learner initialization and the hyper-parameter setting. For the $w_{1,2,3}$ of the $\epsilon$-learner shown in the Eq. (2), at the beginning, we learn the first five segments with the $\epsilon$ sequence $[\frac{1}{4}\tilde{\epsilon}, \frac{1}{2}\tilde{\epsilon}, \tilde{\epsilon}, 2\tilde{\epsilon}, 4\tilde{\epsilon}]$, then track their rewarded $SegErr_i$ and update the parameters $w_{1,2,3}$ using least square regression. We empirically found that this light-weight initialization leads to better index performance compared to the versions with random parameter initialization, and it benefits the exploration of diverse $\epsilon^*$, *i.e.*, leading to the larger variance of the dynamic $\epsilon$ sequence $[\epsilon_1, \ldots, \epsilon_i, \ldots, \epsilon_N]$. As for the hyper-parameter $\rho$ (described in the Section 3.4), we conduct grid search over $\rho \in [0.1, 0.4, 0.7, 1.0]$ on Map an IoT datasets. We found that all the $\rho$s achieve better $N$-*MAE* trade-off (*i.e.*, smaller AUNEC results) than the fixed $\epsilon$ versions. Since the setting $\rho = 0.4$ achieves averagely best results on the two datasets, we set $\rho$ to be $0.4$ for the other datasets.

## G  Dataset Details

Our framework is verified on several widely adopted datasets having different data scales and distributions. *Weblogs* [19, 14, 12] contains about 715M log entries for the requests to a university web server and the keys are log timestamps. *IoT* [14, 12] contains about 26M event entries from different IoT sensors in a building and the keys are recording timestamps. *Map* dataset [19, 14, 9, 12, 21] contains location coordinates of 200M places that are collected around the world from the Open Street Map [25], and the keys are the longitudes of these places. *Lognormal* [12] is a synthetic dataset whose key intervals follow the lognormal distribution: $ln(G_i) \sim \mathcal{N}(\mu_{lg}, \sigma_{lg}^2)$. To simulate the varied data characteristics among different localities. We generate 20M keys with 40 partitions by setting $\mu_{lg} = 1$ and setting $\sigma_{lg}$ with a random number within $[0.1, 1]$ for each partition.

We normalize the positions of stored data into the range $[0, 1]$, and thus the key-position distribution can be modeled as Cumulative Distribution Function (CDF). We plot the CDFs and zoomed-in CDFs of experimental datasets in Figure 7 and Figure 8 respectively, which intuitively illustrate the diversity of the adopted datasets.

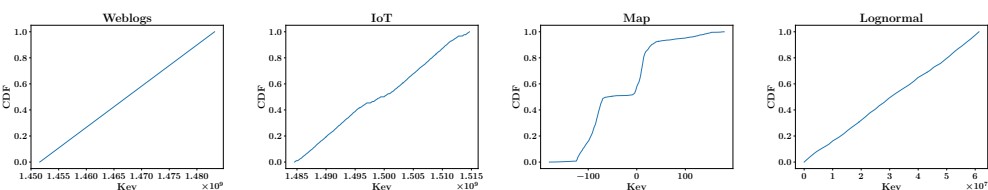

Figure 7: CDFs of adopted datasets.

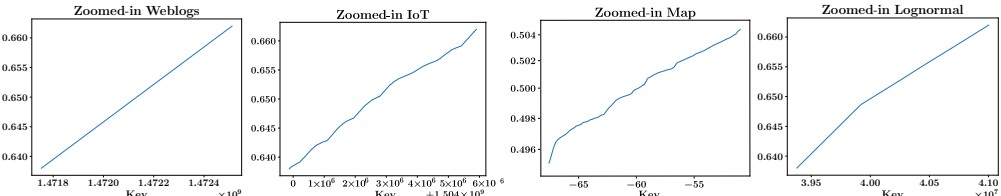

Figure 8: Zoomed-in CDFs of adopted datasets.

## H  Additional Experimental Results

**Overall Index Performance.** For the $N$-*MAE* trade-off improvements and the actual querying efficiency improvements brought by the proposed framework, we illustrate more $N$-*MAE* trade-off curves in Figure 9 and querying time results in Figure 10. We also mark the 99th percentile (P99) latency as the right bar, which is a useful metric in industrial-scale practical systems. Recall that the $N$-*MAE* trade-off curve adequately reflects the *index size* and *querying time*: (1) the *segment size in bytes* and $N$ are only different by a constant factor, e.g., the size of a segment can be 128bit if it consists of two double-precision float parameters (slope and intercept); (2) the querying operation can be done in $O(log(N) + log(|y - \hat{y}|))$ as we mentioned in Section 3.1, thus a better $N$-*MAE* trade-off indicates a better querying efficiency. From these figures, we can see that the dynamic $\epsilon$ versions of all the baseline methods achieve better $N$-*MAE* trade-off and better querying efficiency, verifying the effectiveness and the wide applicability of the proposed framework. Regards the p99 metrics, we can see that the dynamic version achieves comparable or even better P99 results than the static version, showing that the proposed method not only improves the average lookup time, but also has a good robustness. This is because of that our method can effectively adjust $\epsilon$ based on the expected $\tilde{\epsilon}$ and data characteristic, making the $\{\epsilon_i\}$ fluctuated within a moderate range.

**CV as an Indicative Quantity.** The *coefficient of variation* (CV) value, *i.e.*, $CV = \sigma/\mu$, plays an important factor in our bounds to reflect the linearity degree of the data. We have seen that $CV$ is effective to help dynamically adjust $\epsilon$ in our framework as shown in our experiments. Here we explore that *whether the CV value can be an indicative quantity to shed light on what types of data will*

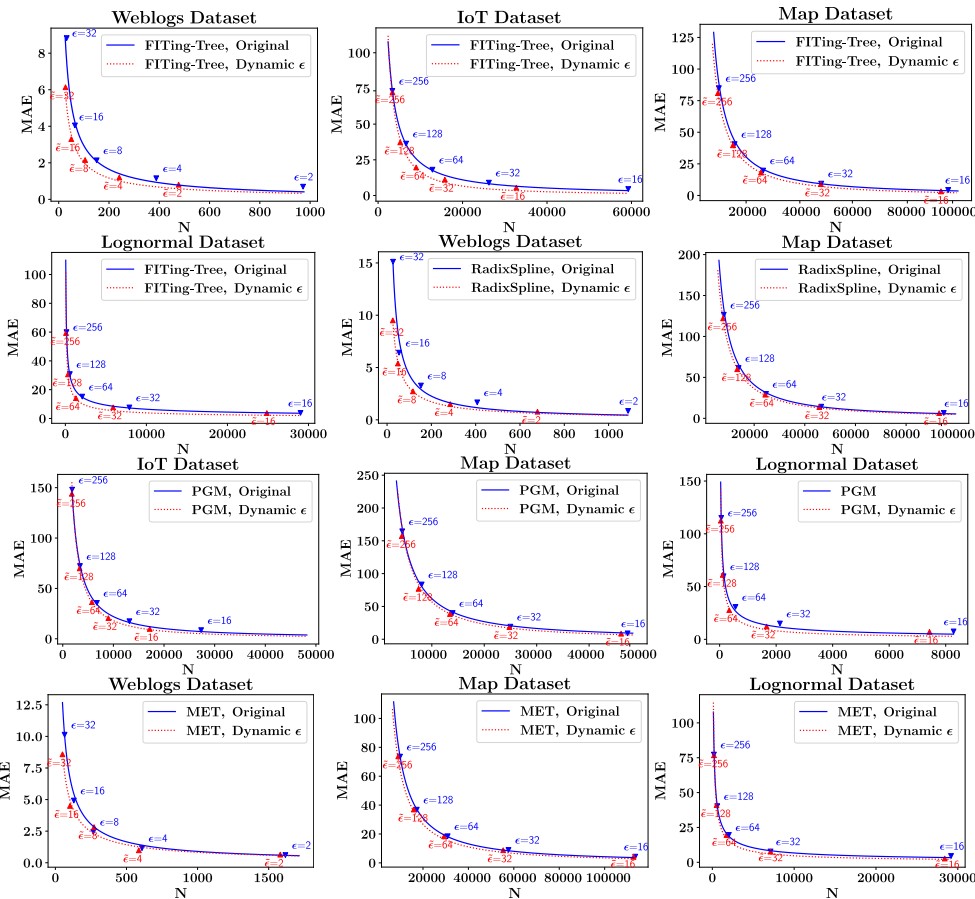

Figure 9: The additional $N$-*MAE* trade-off curves for learned index methods.

*benefit from our dynamic adjustment?* To be specific, we calculate the $CV$ values of the experimental datasets and compare them with the trade-off improvements. The global $CV$ values of IoT, Map, Lognormal and Weblogs are 65.24, 11.12, 0.85, and 0.013 respectively, while their AUNEC improved by 20.71%, 6.47%, 21.89% and 26.96% respectively. With the exception of IoT, the rest of the results show that *the smaller the $CV$ value is, the greater the trade-off improvement of dynamic $\epsilon$ brings*. We find that IoT is a locally linear but globally fluctuant dataset. We then divide the data into 5000 segments and calculate their average $CV$ values. The local $CV$ values of IoT, Map, Lognormal and Weblogs are 0.95, 2.18, 0.63, and 0.005 respectively, which is consistent with the improvement trends. Intuitively, when the local $CV$ value is small, the local data is hard-to-fit with a few linear segments if we adopt an improper $\epsilon$, and we need more fine-grained $\epsilon$ adjustment rather than the fixed setting. Thus we can expect more performance improvements in this case. The calculation of actual $CV$ values of real-world datasets helps to validate our $\epsilon$ analysis based on the $CV$ values, and provides further insight into the scenarios where the proposed method has strong potential to outperform existing methods.

**Ablation Study.** To examine the necessity and the effectiveness of the proposed framework, in Section 4.3, we compare the proposed framework with three dynamic $\epsilon$ variants for the FITing-Tree method. Here we demonstrate the AUNEC relative changes for the Radix-Spline method with the same three variants in Table 4 and similar conclusions can be drawn.

**Theoretical Validation.** In Section 4.4, we show that all the learned index baseline methods learn similar segment slopes on the Map dataset. Here we illustrate the learned slope results on the IoT, Weblogs and Lognormal datasets in Figure 11, which supports the Theorem 2 that the learned segment slopes concentrate on the $1/\mu_i$ with a bounded relative difference.

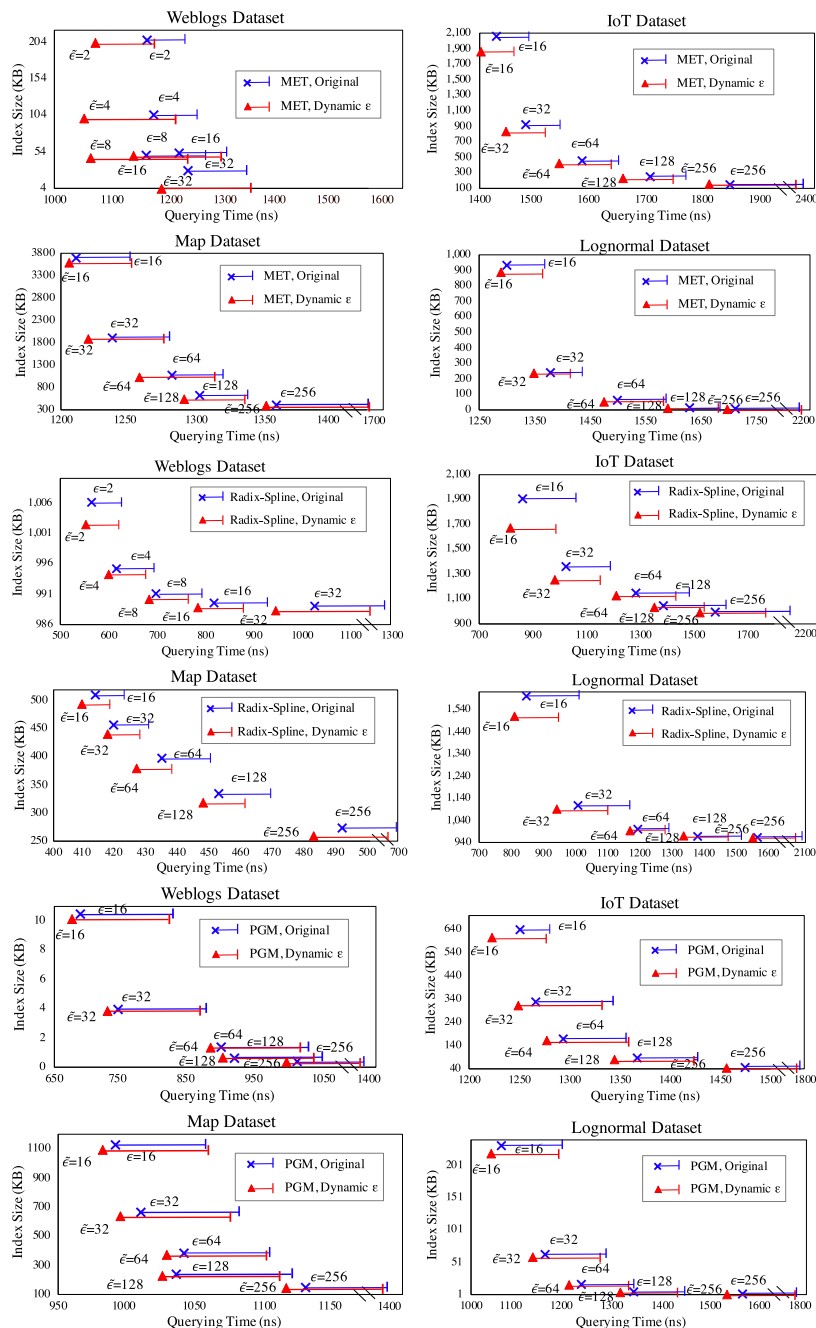

Figure 10: Improvements in terms of querying time for learned index methods with dynamic $\epsilon$.

Besides, for the comparison between the theoretical bounds and the actual $SegErr_i$ of all the adopted learned index methods, we show more results on another two datasets *Gamma* and *Uniform* in Figure 12, where the key intervals of the two datasets follow gamma distribution and uniform distribution respectively. These results show that the MET method gains actual $SegErr_i$ within the bounds, verifying the correctness of the Theorem 1 again. Here all the learned index methods also achieve the same trends, showing that these methods have the same mathematical forms w.r.t. the $SegErr_i$, $\epsilon$ and $\mu/\sigma$, and hence the $\epsilon$-learner can effectively learn the estimator and adaptively choose suitable $\epsilon$.

Table 4: The AUNEC relative changes of dynamic $\epsilon$ variants compared to the Radix-Spline method with the proposed framework.

|  | Random $\epsilon$ | Polynomial Learner | Least Square Learner |
|---|---|---|---|
| Weblogs | +81.23% | +56.20% | -9.56% |
| IoT | +74.78% | +53.28% | +9.81% |
| Map | +60.67% | +7.34% | +0.45% |
| Lognormal | +83.16% | +55.01% | $-11.23\%$ |
| Average | +74.96% | +42.96% | $-2.63\%$ |

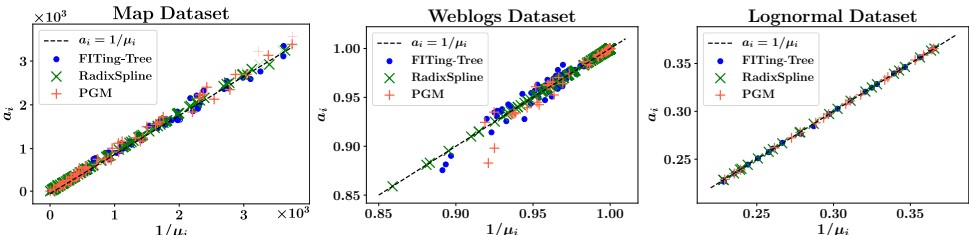

Figure 11: Learned slopes on the IoT, Weblogs and Lognormal datasets.

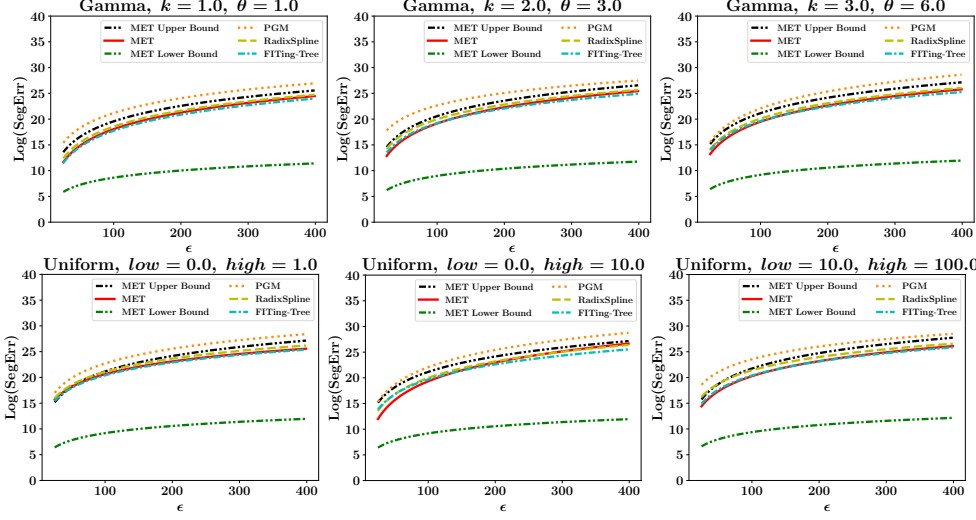

Figure 12: Illustrations of the derived bounds on *Gamma* and *Uniform* datasets.