# OpenReview forum: "Learned Index with Dynamic $\epsilon$"
_NeurIPS.cc/2022/Conference — NeurIPS 2022 Submitted_

### Official Review · Reviewer_ky4v · 2022-07-09

**Rating:** 6
**Confidence:** 3
**Soundness:** 4 excellent
**Presentation:** 4 excellent
**Contribution:** 3 good

**Summary:**

The paper presents a method to learn a variable error bound $\varepsilon_i$ to use when building piecewise linear segments for learned indices for databases.
This additional learned component is based on an estimation function that uses a sample to estimate the mean and the variance of the remaining data which will be covered by the next linear segment.
There is an analysis of the prediction error, making some not-so-strong-but-not-weak assumptions about the data distribution.
The experimental evaluation shows that the additional component greatly improve the space/error trade-off, does not cost too much in terms of indexi building cost, and has similar query costs as existing methods.

**Questions:**

I'm wondering about the appropriateness of using a sample over the whole "leftover" part of the Data to estimate $\mu$ and $\sigma$ (if I'm understanding correctly, a new estimation is taken before building each segment). The estimates are, I assume, the sample mean and sample stddev, but I wonder whether one should be using a more refined model by, e.g., fitting a curve on the sample, or something similar, to get a better approximation of the behavior of the next data points, which should allow to get a better estimates for the parameters, thus for $\varepsilon_i$. Can you please comment on possible different choices for this step, which seems crucial?

**Limitations:**

I would have liked to see more discussion/evaluation of the limitations of the proposed approach. For example, what happens when the assumptions on the distributions are not satisfied.

**Strengths And Weaknesses:**

## Strenghts

* Good idea, and well executed

* Complete experimental evaluation, and convincing.

* Very clear presentation

## Weaknesses

* The whole presentation relies a lot on the MET algorithm, but, as correctly stated and evaluated, the proposed approach can be used with other methods for learning indices. Perhaps the presentation could be reworked to be more generic.

* I found the presentation of the proof sketch not really enlightening, despite being someone who regularly performs similar analysis.

---

> ### Author Response · Authors · 2022-08-02
> **Responses to Reviewer ky4v**
>
> Many thanks for your appreciation and insightful comments! We make the following responses point by point to address your comments:
>
>
> **W1: A more generic presentation not limited on MET.**
>
> Thank you very much for your appreciation and suggestions!
> We introduce MET at some length since it is relatively simple and a good demonstrative method that does not introduce complex learning mechanisms, but conveys the core idea of linear piece-wise approximation for learned index, reducing the effort of understanding for the reader.
> To make the reader easily aware of the generality of the proposed framework, we do link MET to other $\epsilon$ methods as mentioned in line 92, line 223 and Theorem 2.
> Following your suggestion, we will make this clearer by adding more explanations at the end of line 167: "Our framework is general and can be applied to a broad class of $\epsilon$-bounded methods, such as MET, RadixSpline, FITing-Tree and PGM."
>
> **W2: Proof sketch is not enlightening.**
>
> Thank you for your comments!
> We do not over-claim the novelty of the technique tool used by the theoretical analysis, but rather emphasize our novel perspective and the results of the theoretical analysis, i.e., the bounds w.r.t N-MAE trade-off and data characteristics.
> As a comparison, the MET paper mainly focuses on theoretical analysis with validation on synthetic simulation data only, while our proposed framework focuses on improving a broad class of learned index methods, with extensive experimental validation on several real-world datasets.
>
> **Q1: Using a sample over the whole "leftover" part of data.**
>
> Thanks for your detailed comments! Your understanding is right, and using different estimators is feasible.
> However, in terms of efficiency, conducting statistics over the whole remaining data each time would break the one-pass nature of the original learned index methods, and incur a high building overhead. Furthermore, statistics such as $\mu$ and $\sigma$ over the whole remaining data may not effectively characterize the *varied* data distribution due to the simple averaging operation.
> Empirically, we tried using the whole remaining data on some datasets but found it does not work.
> As for the different choices you mentioned, it is possible to use a small part of the remaining (look-ahead) data with different estimators to fit a curve.
> As Section 4.3 shows, we have compared several estimation baselines, including random, polynomial, and least square estimators. The results show the necessity, effectiveness, and efficiency of the proposed estimator.

---

> > ### Comment · Reviewer_ky4v · 2022-08-05
> > **Thank you**
> >
> > Thank your for replying to my questions. My issue with the proof sketch is mostly that it doesn't really give a good intuition: I would say that a proof sketch should certainly highlight the main points of the proof, but also give some intuition of its whole structure, to convince the reader that nothing that was not said is so important (or potentially subtle) that it should have been said.

---

> > > ### Author Response · Authors · 2022-08-07
> > > **Thanks for the response by reviewer ky4v.**
> > >
> > > We sincerely thank the reviewer for your further response and constructive suggestions! We carefully check the existing proof sketch, and in the final version, we will make the proof sketch more structured and convey the basic insights better via the following actions:
> > > - We will form the proof into three sub-sections and make its structure clearer: (1) modeling transformed random walk processes based on $SegErr_i$, $\mu$, $\sigma$ and $\epsilon$; (2) proof of upper bound; and (3) proof of lower bound.
> > > - We will highlight more about how to link the estimated indexing performance with the quantity $SegErr_i$ (in sub-section 1), and give more intuitive discussions about the derived performance bounds (in sub-sections 2 and 3). For example, the upper bound
> > > indicates that the $\epsilon$ has a larger impact on the indexing performance (third power) than the linearity degree of data $\frac{\mu}{\sigma}$ (second power), and the constant $\frac23\sqrt{\frac2\pi}(\frac{5}{3})^{\frac{3}{4}} \approx 0.78$ is tighter than the trivial one with $1$ that corresponds to the case where each data point reaches the largest error $\epsilon$.
> > > - To make the proof more readable and convincing, we will add instructions about which theorems we are using for the omitted steps, and indicate in which lines in the Appendix a more detailed explanation can be referred to.
> > >
> > > Thanks again for your suggestions and appreciation!

---

### Official Review · Reviewer_o36M · 2022-07-11

**Rating:** 6
**Confidence:** 3
**Soundness:** 3 good
**Presentation:** 3 good
**Contribution:** 3 good

**Summary:**

This paper focuses on learning index with dynamic prediction error. The major contribution of this paper is proposing an efficient, pluggable d learned index framework that is adaptive to dynamic prediction error.

**Questions:**

The motivation of dynamic $\epsilon$:

An intuition of dynamic $\epsilon$ is that we can have flexible computational overhead to fit the budget. Does the running time of proposed the framework in query varies with dynamic $\epsilon$? If so, what is the saving if we have relaxed $\epsilon$?

The running time analysis:

1. Is there any extra overhead we should pay to enjoy the adaptivity over different $\epsilon$?

2.  A naive approach for dynamic $\epsilon$  is to maintain a multi-level set of indices, and each is for different $\epsilon$. What is the advantages of the proposed approach to this naive approach in terms of running time and memory?

Robustness to adaptive queries:

As mentioned before, one advantage of dynamic $\epsilon$ is that we can design robust data structures over adversaries. From this point of view, is the current framework allow adaptive queries?  Does the theoretical analysis on the query procedure assume the queries are independent?

**Limitations:**

This paper is trying to solve a problem with practical significance. The current presentation can be improved with answers to the questions in the previous sections. Specifically, a better connection between the dynamic $\epsilon$  setting to practical problems would help the understanding from a broader audience.

**Strengths And Weaknesses:**

Strength:

1. Problem setting: this paper studies an interesting problem in learning to index. In the practical deployment of indexing data structure, the approximation error can vary due to the computation budget. Moreover, the dynamic approximation error would benefit the robustness of the index over the adversary.

2. Method: the proposed method have a solid mathematical foundation. The workflow of theoretical analysis makes sense to the reviewer.

3. Evaluation: the experiment presented in this paper is extensive, making this paper more convincing. A highlight of contribution would be the simulation of different $\epsilon$ patterns in Sec 4.3

---

> ### Author Response · Authors · 2022-08-02
> **Responses to Reviewer o36M, [Part 1, Q1~Q2].**
>
>
> Thank you very much for your appreciation and detailed comments! We make the following responses point by point to address your comments:
>
> **Q1: Does the running time of proposed the framework in query varies with dynamic $\epsilon$?**
>
> Thanks for this insightful question!
> And the answer to your question is yes.
> To quantitatively examine the variation, we indeed have conducted experiments to compare the static and dynamic learned index versions, in terms of their standard deviation (std) and 99th percentile (P99) of query time, which reflects the gained query time distributions.
> Here we list the std and P99 query time for MET and Radix-Spline on Map datasets, more results can be found in the Appendix and they can draw similar conclusions.
> The dynamic versions achieve comparable results on both the two metrics, and even beats the static version in most cases. Note that our method selects $\epsilon$ based on the expected $\tilde{\epsilon}$, making the $\{\epsilon\}$ fluctuated within a moderate range and leading to a good robustness.
>
> | $\epsilon$ | Metric |   MET   | Dynamic Version | Radix-Spline | Dynamic Version |
> |------------|--------|---------|-----------------|--------------|-----------------|
> | 16         | std    |  494.89 |          494.05 |       457.61 |          145.08 |
> |            | P99    | 2042.48 |         2015.97 |       708.51 |          701.31 |
> | 32         | std    |  478.33 |          465.28 |       381.04 |          120.69 |
> |            | P99    | 1976.36 |         1941.49 |       584.44 |          581.55 |
> | 64         | std    |  455.18 |          444.64 |       308.14 |          106.51 |
> |            | P99    | 1945.65 |         1913.47 |       527.74 |          527.60 |
> | 128        | std    |  429.28 |          420.82 |       201.12 |           95.91 |
> |            | P99    | 1911.05 |         1903.97 |       491.48 |          500.82 |
> | 256        | std    |  404.24 |          403.77 |       142.49 |           86.71 |
> |            | P99    | 1919.15 |         1926.61 |       471.65 |          466.26 |
>
>
> **Q2.1: For running time, is there any extra overhead we should pay?**
>
> Thanks for your question, we clarify that the extra overhead we paid is in the building time rather than the querying time, and our framework is efficient as it retains the online learning manner with the same complexity as the original methods (both in $O(\mathcal{D})$).
> The proposed technique slightly increases the the total building time, which includes (1) the data look-ahead process, (2) the training process of $\epsilon$-Learner (i.e., updating its parameters), and (3) the inference process of $\epsilon$-Learner (i.e., predicting suitable $\epsilon$ for remaining data. These steps are both light-weight and fast, and lead to efficient learning as the experiments (Table 2) shown.
>
> **Q2.2: Mutli-level set of indices.**
>
> Thanks for your insightful question. Actually, the adopted baselines FITing-Tree and PGM can be regarded specific instances of your mentioned multi-level set, as they adopt tree-based data layout. FITing-Tree organizes the learned segments with a B+-Tree and PGM organizes the segments with a multi-level recursive structure. With the proposed framework, these methods maintain multi-level set of indices, and each is for different $\epsilon$. The key process is the adjustment mechanism of $\epsilon$ for different segments, which we investigated in the Ablation Study (Sec. 4.3). Compared with several baselines including random, polynomial, and least square estimators, we find that the results verify the necessity, effectiveness, and efficiency of the proposed method.

---

> ### Author Response · Authors · 2022-08-02
> **Responses to Reviewer o36M, [Part 2, Q3 & Connection to practical problems].**
>
> **Q3: Robustness to adversaries.**
>
> Thank you for your question! Let's discuss your mentioned adversaries into malicious and non-malicious cases, i.e., whether someone actively wants to harm the indexing performance.
> In database indexing applications, malicious adversaries are usually difficult to succeed because database administrators will manage the permissions and carefully design security mechanisms (advanced attacks are out of our scope).
>
> For the non-malicious adversaries, we assume that most audience are interested in the case where datasets intrinsically include (drastically) varied data distribution.
> In fact, we have taken this adversarial distribution shift scenario into account in the design of the proposed method.
> We propose to first probe the distribution properties of only a small fraction of the remaining data (via look-ahead data), and then adapt and adjust the $\epsilon$ accordingly.
> This timely adjustment makes the learned index more robust, which is not only supported by our theoretical analysis, but also the effective improvements of space-time performance on real-world datasets as our experiments show.
> For example, the CDF visualization of the Map dataset (Fig.7 in Appendix) shows that it has a fairly shifted distribution across different data localities.
>
> **"Better connection between dynamic $\epsilon$ setting to practical problems."**
>
> Thanks for your suggestion and we will add more discussion about this in the final version.
> Database indexing is very widely used in real-world applications, and the need to change $\epsilon$, that is, the scenario where there is varied local data distribution, is also very common.
> For example, the adopted Weblog dataset has typically non-linear temporal patterns caused by online campus transactions such as class schedule arrangements, weekends and holidays.
> The IoT dataset contains more complex temporal patterns since the diverse data sources such as motions and doors are recorded with irregular time intervals and diverse behaviors of peoples.

---

### Official Review · Reviewer_X1hc · 2022-07-11

**Rating:** 5
**Confidence:** 3
**Soundness:** 2 fair
**Presentation:** 2 fair
**Contribution:** 2 fair

**Summary:**

This paper proposes a pluggable method that learns a dynamic prediction error bound \epsilon for some error-bounded learned indexes. With the proposed method, the learned index can make a better trade-off between space and time. Experiments on real world data show the efficacy of applying the proposed method on existing learned indexes.

**Questions:**

1. How does your dynamic \epsilon framework with SOTA \epsilon-bounded learned indexes compare to these works in terms of indexing performance?
2. When the distribution shifts, does the learned \epsilon still fit?

**Limitations:**

There is no negative societal impact.

**Strengths And Weaknesses:**

strength:
1. The paper is well-organized and the motivation is easy to follow.
2. The idea is easy to understand and interesting.

weakness:
1. In the motivation, the authors explain why the parameter \epsilon affects the N-MAE trade-off, thus affects the performance of the learned index in terms of the index size and indexing speed. It not clear that how much the parameter \epsilon affects the indexing performance compared to other aspects (e.g., data layout) of the learned index. The motivation should provide more preliminary results on different \epsilon to show the impact is significant and learning \epsilon is worthy, since the Figure 3 and Figure 10 show the maximum improvement on the index size and query time is only about 1MB and less 100ns.
2. The \epsilon-Learner requires users input an expected \epsilon before it starts to learn the \epsilon. This requirement is not reasonable enough which conflicts with the semantics of the dynamic \epsilon framework. The prediction errors which related to \epsilon only indirectly reflects the indexes' performance. To obtain a good performance (faster indexing and smaller index size), users still have to try different \epsilon to know the quantitive relations between \epsilon and indexing speed or size. For example, if users expect that the query time of FITing-Tree is less than 900 ns, they need to try different \epsilon until they find the fact that \epsilon should be smaller than 16 (Figure 3).
3. Recent works, e.g., ALEX, LIPP, have achieved a better performance compared to that of PGM-index, FITing-Tree. These works doesn't need a \epsilon to bound and have shown that the performance of learned indexes can be significantly improved with a better data layout. It seems not necessary to for the more advanced learned index to include the dynamic \epsilon.
4. In the results of index building cost, the paper claims the extra cost is only paied once. However, the paper lacks the experiments of the impact of updates on \epsilon-Learner.

---

> ### Author Response · Authors · 2022-08-02
> **Responses to Reviewer X1hc, [Part 1, W1~W2].**
>
> Many thanks for the insightful and helpful comments by the reviewer! We make the following responses point by point to address your comments:
>
> **W1: Significance of the performance improvements with dynamic $\epsilon$.**
>
> Many thanks for your suggestion to provide more results to show the significance of performance improvements with dynamic $\epsilon$.
> Although the maximum *absolute* improvements in the index size and query time are about 1MB and 100ns respectively, the *relative* improvements are significant by up to about $82$% and $12.9$% respectively (FITing-Tree on IoT, $\epsilon=16$, Fig. 3).
> Note that the reported query time corresponds *a single query*, and the proposed algorithm framework *retains the online learning manner* (with linear complexity). This means that we can scale to very large datasets and achieve considerable improvements in terms of the total query time.
>
> As for your mentioned effect of $\epsilon$ on data layout, we take it into account in the left term $log(N)$ of the complexity analysis in line 122, since most considered $\epsilon$-bounded learned index methods adopt tree-based data layouts and require finding the specific segment responsible for a given query $x$ in $log(N)$ time. To summarize, adjusting $\epsilon$ will lead to different learned $N$ (trees with different heights and depths), and indirectly affect the overall query time.
>
>
> **W2: Usability of $\epsilon$.**
>
> Thanks for your detailed comments. We discussed the usability of $\epsilon$ in Section 3.2 (line 168 ~ 177).
> Specifically, we note that the proposed pluggable framework provides users the same interface as the ones used by original learned index methods.
> Compared to original $\epsilon$-bounded learned index methods, we *do not* add any additional cost to the users' experience.
> Users can smoothly and painlessly use our framework with given  $\tilde{\epsilon}$ just as they use the original methods with given $\epsilon$.
> We note that the $\{\epsilon\}$ is an easy-to-set quantity for users and is easier to estimate than the other quantities such as querying time and index size, which are dependent on specific algorithms, data layouts, implementations and experimental platforms.
>
> As for your mentioned hard indexing performance limitation case, we also provide discussion about how to deal with it via inheriting the abilities of the existing work PGM in Appendix E.
> Specifically, we observe that PGM introduced a multi-criteria variant that auto-tunes itself with pre-defined hard indexing requirements from users, i.e., the min-time mode and min-size mode discussed in Section 7.5 of PGM paper.
> Our proposed framework is pluggable and still valid when using the PGM variant to handle the hard time or size requirements.
> Here we take the hard size requirement as an instance:
> given a space constraint, the multi-criteria PGM proposes to iteratively estimate the relationship between $\epsilon$ and $size$ with a learnable function $size(\epsilon)=a\epsilon^{-b}$, and automatically outputs the index that minimizes its query time via different estimated $\epsilon$s. Given a size requirement, we can just do the same thing in dynamic $\epsilon$ scene by setting our $\tilde{\epsilon}$ as $\epsilon$ estimated by the original PGM method.

---

> ### Author Response · Authors · 2022-08-02
> **Responses to Reviewer X1hc, [Part 2, W3~Q2].**
>
> **W3: Learned index optimization from the view of $\epsilon$ v.s. data layout.**
>
> Thanks for your comments about the non-$\epsilon$ bounded methods. These two performance optimization perspectives are relatively orthogonal.
> Both types of the learned index methods have their pros and cons:
> - data layout based approaches such as ALEX do achieve better indexing performance in dynamic scenarios, but it pays a **larger index size overhead** because of the introduced gap insertion technique (reserving empty space for possibly inserted data).
> Empirically, we examine the performance of ALEX with our experimental settings and find that it gains comparable query time and larger index size than learned index with dynamic $\epsilon$ as following results show, where the q_time and size are in ns and KB respectively. We adopt the default hyper-parameters of ALEX and the $\tilde{\epsilon}$ is 4, 32, 64 and 32 in these four datasets respectively for RadixSpline.
>
> |                                 | Weblogs|        |  IoT       |        |  Map |        | Lognormal |    |
> |----------------------- |---------  |------|--------|------|--------|------|--------|-------|
> |                                 | q_time   | size | q_time  | size | q_time | size | q_time |  size |
> | ALEX                            | 713     | 1428 | 915    | 1632 | 521    | 582  | 962    |  1321 |
> | RadixSpline, Dynamic $\epsilon$ | 602     | 995  | 974    | 1275 | 428    | 373  | 937    |  1085 |
>
> - For the $\epsilon$-bounded methods, the most important advantage they provide is the **worst-case guarantee** in each segment. This property is fairly valuable in many realistic indexing applications such as financial databases and on-device intelligence. Also note that our approach still maintains comparable 99th percentile (P99) performance compared to static $\epsilon$ baselines as shown in Fig.3 and Fig.10.
>
> - Moreover, our novel framework can be regarded as an automatic method that determines hyper-parameters ($\epsilon$) according to the varied local properties of the data. Although the data layout optimization goes beyond our scope, *our insight can also contribute to the ALEX approach*, since it also introduces hyper-parameters in local linear segments learning, the lower and upper density limits on each gapped array: $d_l,d_u \in (0,1]$. The authors empirically set them to be 0.6 and 0.8 respectively. However, different gapped arrays may gain better performance with different density limitations, since the data distribution can be varied across different localities, as we have shown in experiments.
>
> **W4: Experiments of the impact of updates on $\epsilon$-Learner.**
>
> We clarify that the *index building process includes the update process of $\epsilon$-Learner*. Actually, the experimental results in Table 2 indicate total time increments including (1) the data look-ahead process, (2) the training process of $\epsilon$-Learner (i.e., updating its parameters), and (3) the inference process of $\epsilon$-Learner (i.e., predicting suitable $\epsilon$ for remaining data. These steps are both light-weight and fast, and they lead to efficient learning as the experiments shown.
>
>
> **Q1: Performance comparison between $\epsilon$ bounded methods and your mentioned methods based on data layout optimization.**
>
> Please see the results and discussion in the above response to W3.
>
> **Q2: When the distribution shifts, does the learned $\epsilon$ still fit?**
>
> In fact, we have taken this distribution drift scenario into account in the design of the proposed method. We propose to first probe the distribution properties of only a small fraction of the remaining data (via look-ahead data), and then adapt and adjust the $\epsilon$ accordingly. This timely adjustment makes the learned index more robust, which is not only supported by our theoretical analysis, but also the effective learning on real-world datasets as our experiments show. For example, the CDF visualization of the Map dataset (Fig.7 in Appendix) shows that it has a fairly shifted distribution across different data localities.
>
> We sincerely thank that the reviewer consider increasing the rating scores if our clarification addressed your comments. Please let us know if you have any additional questions or suggestions. We are happy to engage more and address them fully!

---

> ### Author Response · Authors · 2022-08-08
> **Thank you again for your insightful and helpful comments!**
>
> Dear Reviewer X1hc,
>
> Thank you again for your time and efforts in reviewing our paper, as well as the insightful and helpful review! We carefully read and responded to all your comments. Does our response address your comments? We would appreciate the opportunity to engage further if needed.

---

> ### Comment · Reviewer_X1hc · 2022-08-08
> **The motivation is weak and experiments are not solid - summary**
>
> Thank authors for the response.
> Here is the conclusion of weakness of this paper after reading the responses:
> 1. The motivation is still too weak for a top conference. Submitting this work to NeurIPS does not mean you do not need to resolve a practical database problem.
> - The family of building learned using epsilon seems already far from the mainstream of the learned index research. This method is not used in the recent high-performance learned index such as [1][2][3].
> - Index size (only the piece-wise linear models) is never a big concern. Would a few MB (2 MB) index size cause a huge overhead for the memory? It is negligible even for severs from 10 years ago.
> 2. Throughput or latency is one of the most important metric in this fields. And this paper is far from the SOTA methods. The new experimental results for throughput seems problematic and not acceptable. And more experimental results are expected.
> - The Alex only has a around 70-90 ns running time. And the reported time is 500 - 900 ns. Is it due to different setups of benchmark? As ALEX has a well open-sourced benchmark, it won't be hard to reproduce. I would consider reduce the score if the experimental results are problematic, as the throughput is important for the learned index.
> - Even if the experiments are correct, the claimed improvement of throughput (12.9%) is not consistent and sometimes (the IoT dataset) the throughput is degraded.
> - The SOTA methods [1][2][3] have improved the query latency to around 30ns (30x faster than this paper's results) for lognormal and longitudes datasets. I do not think 937ns is competitive.
> 3. A user pre-defined epsilon does not make sense to me. If this paper claim the dynamic epsilon as the major contribution, it's kind of awkward if every epsilon should be tuned from a pre-defined value. This problem was raised by the reviewers from ICLR as well.
>
> I'll leave the detailed comments in the following chat box.

---

> ### Comment · Reviewer_X1hc · 2022-08-08
> **The motivation is weak and experiments are not solid - details**
>
> Here are the detailed comments after reading the responses.
>
> 1. For the replies to W1, I still expect to see strong support to show the significance of adjusting the hyperparameters. The reason is that existing SOTA learned indexes (such as ALEX [1], LIPP [2], NFL [3]) have achieved great performance, which proves that designing new structures/strategies can bring much more improvements than that of adjusting hyperparameters. From this point, the motivation of this paper might be weak.
>
> As for the relative improvements, I expect to conduct a scalability experiment (like Figure 12 in [1]) to scale your methods to large datasets. The experiments should vary from different hyperparameters as well as data layout and show which part brings the major improvements. This is because the index performance is also determined by the data layout. For example, the query performance on the tree-based index depends on the tree height and the local search efficiency. The \epsilon in PGM only affects the number of segments in each index's level and the local search efficiency. The tree height depends on another hyperparameter named 'epsilon_recursive'. Thus, I have questions about "scale to very large datasets and achieve considerable improvements".
>
> 2. For the replies to W3, here are some questions about new experimental results. According to the Figure 9 (a) in [1], the throughput of ALEX on the read-only workload of Map is roughly about 11 million ops/sec, which is about 1,000,000,000 / 11,000,000 \approx 90 nanosecond/op. The difference between ALEX's result (90ns) and your result (521 ns) is quite significant which is not acceptable. It seems that ALEX's performance significantly degrades by an order of magnitude.  The same problems exist in the results on the Lognormal dataset although the variance in your setup is different. I wonder to know the detailed reasons for the huge performance drop and detailed experimental setup.
>
> As for the fact that ALEX introduces a larger index size overhead, I don't think it is a significant problem for learned index. The reason is that the reported index size (which actually is the overall size of linear models) is only about 1~2 MB for 200 million double keys, this is much smaller than the common memory capacity in modern computers (at least 2 GB for laptops).
>
> 3. For the replies to W4 and Q4, this can be a minor issue. What I ask is about the impact of inserting new keys on the learned \epsilon. I agree that the training and inference process can be seen as a kind of updating scenario. However, the practical updating problem can be much more complex, which involves in-domain/out-of-domain updates where the inserted keys are in/out of the loading ranges (like Figure 12 (b) in [1]), or distribution shifts where loading keys and inserted keys are sampled from different CDF (like Figure 12 in [4]).
>
> [1] Jialin Ding, Umar Farooq Minhas, Jia Yu, Chi Wang, Jaeyoung Do, Yinan Li, Hantian Zhang, Badrish Chandramouli, Johannes Gehrke, Donald Kossmann, David B. Lomet, Tim Kraska. ALEX: An Updatable Adaptive Learned Index. SIGMOD Conference, 2020, 969-984.
>
> [2] Jiacheng Wu, Yong Zhang, Shimin Chen, Yu Chen, Jin Wang, Chunxiao Xing. Updatable Learned Index with Precise Positions. Proc. VLDB Endow. 14(8), 1276-1288 (2021).
>
> [3] Shangyu Wu, Yufei Cui, Jinghuan Yu, Xuan Sun, Tei-Wei Kuo, Chun Jason Xue. NFL: Robust Learned Index via Distribution Transformation.
>
> [4] Chaichon Wongkham, Baotong Lu, Chris Liu, Zhicong Zhong, Eric Lo, Tianzheng Wang. Are Updatable Learned Indexes Ready?

---

> ### Author Response · Authors · 2022-08-09
> **New Responses to Reviewer X1hc, [Part 1, New W1].**
>
>
>
> Thanks for the responses. We make the following responses to your newly summarized weaknesses:
>
> **New-W1, "weak motivation"**
> - "Submitting this work to NeurIPS does not mean you do not need to resolve a practical database problem."
>     - We have indeed conducted extensive experiments on practical database datasets (Weblogs, IoT, Lognormal, Map) that are widely adopted in numerous learned index baselines, and the experiments are also appreciated by the other two reviewers o36M and ky4v such as ""complete", "extensive" and "convincing"".
>     - Our framework mainly targets on the $\epsilon$-based methods, which can meet the worst-case requirement in practical application. This kind of practical application aims to ensure the worst performance of the "error correction" stage in the two-stage query method.
>     - As in our response to Reviewer o36M, the dynamic epsilon setting can be connected to some certain practical problems. Database indexing is very widely used in real-world applications, and the need to change $\epsilon$, that is, the scenario where there is varied local data distribution, is also very common. For example, the adopted Weblog dataset has typically non-linear temporal patterns caused by online campus transactions such as class schedule arrangements, weekends and holidays. The IoT dataset contains more complex temporal patterns since the diverse data sources such as motions and doors are recorded with irregular time intervals and diverse behaviors of peoples.
>     - We have always stated that existing $\epsilon$-based methods will not lose their original features when applied to our framework, which means that index methods which can handle data deletion and updating can still handle these cases when using the proposed dynamic $\epsilon$, such as PGM-Index.
>
> - "The family of building learned using epsilon seems already far from the mainstream of the learned index research."
> We would like to clarify your misunderstandings about the $\epsilon$-bounded methods as follows:
> 	- $\epsilon$-bounded learned index methods are useful for many practical database problem requiring worst-case guarantees, and they are still receiving widespread interest and ongoing development such as works in very recently years [1,2]. Thanks for mentioning more index works, and we will add them to the related work section with comparison to appreciate their effort on indexing performance improvements.
> 	- As for your mentioned performance difference between the $\epsilon$ methods and data-layout optimization based works, we indeed conduct fair comparison and our results show that they still achieve comparable performance, and some supports can be found in existing works, please see more details in the following response to W2.
>
> - "Index size is never a big concern."
> 	- Index size is still very important even with the large amount of memory available today, as shown by the fact that almost all related learned index works examine this metric in their experiments. It is worth noting that index size can have a significant impact on latency indirectly, since the smaller the index, the more it can fit into the hierarchical high speed CPU caches, and thus increase the hit rates to speed up the overall indexing performance [3,4].
> 	- Moreover, in some practical applications, the available memory can be relative small such as cases where users need to build index from multiple keys of the data, and need to use index on IoT devices for edge computation.
>
>
> References
>
> [1] Why Are Learned Indexes So Effective? 2021, ICML.
>
> [2] PLEX: Towards Practical Learned Indexing, 2021, Applied AI for Database Systems and Applications.
>
> [3] CARMI: A Cache-Aware Learned Index with a Cost-based Construction Algorithm, 2022
>
> [4] COLIN: A Cache-Conscious Dynamic Learned Index with High Read/Write Performance, 2021, Journal of Computer Science and Technology

---

> > ### Comment · Reviewer_X1hc · 2022-08-09
> > **The responses partially addressed my concerns.**
> >
> > Thanks for the prompt responses. And I should apologize for my late comments before.
> >
> > 1. For the motivation part, the worst case guarantee is useful. I focused mainly on the efficiency-oriented learned index rather than the analysis part. Thanks for noticing this point.
> > 2. For the index size problem, hitting the CPU cache makes sense. But I guess more keys to be placed in the CPU cache would be more useful for efficiency than model being placed in cache, due to my experience in the learned index.
> > 3. The numbers in experiments are kind of convincing if considering the index size.
> > 4. I am still wondering the flexibility of this method, with the user pre-defined epsilon.
> > ------
> > Therefore, I will raise my score, and lower my confidence.

---

> ### Author Response · Authors · 2022-08-09
> **New Responses to Reviewer X1hc, [Part 2, New W2~W3].**
>
>
> **New-W2, "Experiments are not solid"**
>
> We would like to clarify your misunderstandings about the solidness of the experiments. Please see the detailed responses below:
> - We note that ALEX did not directly compare its performance with the line of $\epsilon$-bounded works in their paper. For fair comparison, we have been adopted the official ALEX codes to conduct the experiments with default hyper-parameters. After careful checking, we find that the noticeable difference is the experimental hardware. The authors of ALEX mentioned that they conduct experiments using Intel Core i9-9900K CPU 3.6GHz with 16Mb smart cache, which has more advanced architecture, higher main frequency and stronger cache than ours, Intel(R) Xeon(R) Platinum 8163 CPU @ 2.50GHz with 32K L1 Cache and 1M L2 Cache.
> - Furthermore, we can get some corroboration from the results reported in the papers of these two line works that the $\epsilon$-based methods are still competitive. For example, the query latency improvement of ALEX over B-Tree on the lognormal dataset is not as high as that of Radix-Spline over B-Tree. Judging from the histograms in the papers (see Figure 9a in ALEX, Figure3 in Radix-Spline), the improvement of ALEX is no more than 3 times , and the improvement of Radix-Spline is at least 5 times.
> - Also note that comparisons between adopted baselines in the same hardware environment are fair. Furthermore, although the absolute times are different, the relative performance differences we report are consistent with those in these papers, supporting the soundness of our experiments. For example, in our experiment on Weblogs, the reported query latency is on the order of 1000ns, which is quite similar to those in PGM and FITing-Tree (see Figure3 in ours, Figure8a in FITing-Tree and Figure 6 in PGM).
> - As for the inconsistency you mentioned (point 2.2), which is a misunderstanding, since it is for a comparison between the results of two different experiment scenarios. The 12.9% improvement in our response refers to the improvement of the dynamic epsilon version relative to the static one. The degradation you mentioned refers to the fact that the query performance of Radix-spline with dynamic epsilon is degraded on IoT compared with Alex, while it is even better than Alex on the other three datasets.
>
>
> **New-W3, "A user pre-defined epsilon does not make sense to me"**
>
> We always emphasize that the pre-defined $\epsilon$ dose not add any additional cost to the users' experience just as they use the original $\epsilon$-based methods, which reflects the user's preference between index size and query performance. We dynamically and adaptively adjust the $\epsilon$ around the pre-defined one in order to achieve better performance than the versions adopted fixed $\epsilon$, meanwhile keeping the benefits and usability of $\epsilon$ to users, as it is easy-to-set, intuitive quantity with provable worst-case guarantee.

---

### Meta-Review · Area_Chair_19D1 · 2022-08-27

**Recommendation:** Reject
**Confidence:** Less certain

**Metareview:**

All of the reviewers recommended acceptance, but the support was lukewarm, with the maximum score being “Weak Accept”. There were concerns about the limited applicability of the proposed method and lack of clarity of some of the arguments in the paper. Although the reviewers appreciated the mathematical foundations and experimental results, the negatives outweighed the positives.

**Award:**

No

---

### Decision · Program_Chairs · 2022-09-14

Reject